# HShare: Fast LLM Decoding by Hierarchical Key-Value Sharing

**Huaijin Wu** [12†]**, Lianqiang Li** [2†]**, Hantao Huang** [2]**, Yi Tu** [1]**, Jihang Zhang** [2]**,
Minghui Yu** [2]**, Junchi Yan** [1*]

[1] Shanghai Jiao Tong University, Shanghai, China
[2] ByteDance Inc.

`{whj1201,yanjunchi}@sjtu.edu.cn    tuyi.sjtu2023@gmail.com`
`{lilianqiang,huanghantao,zhangjihang,yuminghui.exp}@bytedance.com`

## Abstract

The frequent retrieval of Key-Value (KV) cache data has emerged as a significant factor contributing to the inefficiency of the inference process in large language models. Previous research has demonstrated that a small subset of critical KV cache tokens largely influences attention outcomes, leading to methods that either employ fixed sparsity patterns or dynamically select critical tokens based on the query. While dynamic sparse patterns have proven to be more effective, they introduce significant computational overhead, as critical tokens must be reselected for each self-attention computation. In this paper, we reveal substantial similarities in KV cache token criticality across neighboring queries, layers, and heads. Motivated by this insight, we propose HShare, a hierarchical KV sharing framework. HShare facilitates the sharing of critical KV cache token indices across layers, heads, and queries, which significantly reduces the computational overhead associated with query-aware dynamic token sparsity. In addition, we introduce a greedy algorithm that dynamically determines the optimal layer-level and head-level sharing configuration for the decoding phase. We evaluate the effectiveness and efficiency of HShare across various tasks using three models: LLaMA2-7b, LLaMA3-70b, and Mistral-7b. Experimental results demonstrate that HShare achieves competitive accuracy with different sharing ratios, while delivering up to an $8.6\times$ speedup in self-attention operations and a $2.7\times$ improvement in end-to-end throughput compared with FlashAttention2 and GPT-fast respectively. The source code is publicly available at `https://github.com/wuhuaijin/HShare`.

## 1 Introduction

The development of Large Language Models (LLMs), like the GPT and LLaMA series (OpenAI, 2024; Touvron et al., 2023), has marked a major breakthrough in artificial intelligence, dramatically improving performance in natural language processing tasks such as translation (Zhu et al., 2023; Pal et al., 2024) and summarization (Zhang et al., 2023; Liu et al., 2023). However, in long-context scenarios, such as multi-turn dialogues (Yi et al., 2024; Teng et al., 2024), document-based question answering (Abdel-Nabi et al., 2023; Rasool et al., 2024), and code completion (Yang et al., 2023; Eghbali & Pradel, 2024), LLMs face significant speed challenges with token-by-token decoding. A key factor contributing to this slowdown is the handling of the Key-Value (KV) memory cache since as the context length expands, the size of this KV cache grows correspondingly, resulting in longer access times and increased memory overhead.

Existing works have introduced several approaches to address this issue. Since it has been demonstrated that a small portion of the tokens can dominate the accuracy of token generation, many works choose to only load these critical tokens into the KV cache to reduce the inference latency while maintaining accuracy. Among them, StreamingLLM (Xiao et al., 2023) treats the initial tokens (also referred to as *sink* tokens) and the recent tokens as critical tokens and performs sparsity in a fixed pattern.

---

*Correspondence author. † equal contribution. Work was partly supported by NSFC (62222607), Shanghai Municipal Science and Technology Major Project (2021SHZDZX0102). Huaijin Wu's work was completed during her internship at ByteDance.

H2O (Zhang et al., 2024b) introduces a greedy policy that dynamically retains a balance of recent and heavy tokens. These two methods alleviate both storage and retrieval pressures through KV cache eviction. However, they tend to lose a significant amount of historical information, leading to a substantial decline in performance as their sparsity increases.

On the other hand, Quest (Tang et al., 2024) points out the criticality of tokens can change with different query tokens. To this end, Quest introduces a query-aware token sparsity algorithm, which uses the maximum and minimum values of each hidden dimension at page granularity to measure the query-aware criticality. Similarly, DoubleSparse (DS) (Yang et al., 2024b) proposes to select critical tokens dynamically by calculating and sorting approximate attention weights with important channels only. Although these two methods only load critical tokens, they retain all the KV cache, allowing them to enhance speed while effectively maintaining accuracy. However, for Quest and DS, the requirement evaluation for each dynamic selection introduces computational overhead, which results in a significant drawback of these approaches. Take DS as an example, with a sequence length of 2k and a batch size of 8, the process of selecting important tokens consumes nearly 50% of the total runtime in their token sparse attention. The comparison of different methods is shown in Tab. 1.

In this work, we further observe that the criticality of KV cache tokens demonstrates significant similarity across different layers, heads within the same layer, and between adjacent queries. Building on this observation and to mitigate the computational overhead associated with the dynamic selection of critical tokens mentioned above, we propose HShare, a hierarchical key-value sharing framework that operates at three levels: layers, heads, and queries. Specifically, we introduce an algorithm to design a sharing configuration for the layer and head levels based on the similarity of critical KV cache token indices. To ensure the sharing accuracy, we compute the corresponding configuration online for each batch of samples after the prefilling phase, the configuration is then applied to the entire decoding phase. This online calculation introduces only a minor increase in the prefill phase's time without incurring any additional overhead during the decode phase. At the query level, we simply share critical token indices between adjacent queries, as their proximity results in higher similarity. By retaining the full KV cache, selectively loading only the critical portions, and implementing hierarchical multi-level sharing, HShare maintains accuracy while reducing the time spent selecting critical KV tokens, significantly improving decoding latency compared to existing methods.

We use LLaMA2-7b-chat (Touvron et al., 2023), LLaMA3-70b (Dubey et al., 2024), and Mistral-7b (Jiang et al., 2023a) to evaluate the accuracy of HShare across three benchmarks: GSM8K (Cobbe et al., 2021), COQA (Reddy et al., 2019), and LongBench (Bai et al., 2023). Experimental results demonstrate that, under the same token sparsity, HShare can maintain accuracy with different sharing ratios, offering competitive performance to state-of-the-art (SOTA) methods. Furthermore, we evaluate the efficiency of HShare and the results show that HShare can achieve up to $8.6\times$ self-attention latency reduction compared with FlashAttention-2 (Dao, 2023) and $2.7\times$ throughput improvement in end-to-end inference compared with GPT-fast (PyTorch, 2023). In summary, **the contributions of this paper are:**

- We systematically analyze the criticality of KV cache tokens across all the levels: different layers, different heads within the same layer, and adjacent queries. Our empirical findings on LLaMA2-7b-chat show that all three levels exhibit substantial similarity.

- We propose a hierarchical critical KV token indices sharing framework with a greedy algorithm to dynamically determine the sharing configuration. To the best of our knowledge, this is the first work to introduce the concept of sharing critical KV cache token indices.

- We evaluate HShare in terms of accuracy and efficiency, with results showing that it maintains model performance with different sharing ratios while achieving up to $8.6\times$ reduction in self-attention latency and $2.7\times$ improvement in inference throughput.

## 2 RELATED WORK

### 2.1 LONG-CONTEXT MODEL

Recently, significant efforts have been made to extend the context windows of LLM, both in academia (e.g., LongChat (Li et al., 2023a), Yarn-LLaMA-2 (Peng et al., 2023)) and industry (e.g., GPT-4 Turbo, which supports up to 128K tokens (OpenAI, 2024)). These extended-context LLMs excel in tasks such as multi-turn conversation comprehension and meeting summarization by providing

Table 1: Comparison of the selection of critical tokens in different token sparsity methods.

| Methods | Selection mode | Efficiency | Memory cost | Accuracy |
|---|---|---|---|---|
| StreamingLLM (Zhang et al., 2024b) | Fixed pattern | High | Low | Low |
| H2O (Xiao et al., 2023) | Dynamic+Local | High | Low | Low |
| Quest (Tang et al., 2024) | Dynamic | Low | High | Median |
| DS (Yang et al., 2024b) | Dynamic | Median | High | High |
| **HShare (Ours)** | Dynamic+Sharing | High | High | High |

enhanced in-context learning and improved performance on complex reasoning tasks. However, this advancement comes with notable trade-offs, including increased computational demands, higher memory usage, and greater bandwidth requirements, leading to elevated costs and longer per-token latencies (Pan et al., 2024; Chen et al., 2024b). For example, Pan et al. (2024) utilizes host SSD memory to alleviate the long sequence KV cache memory requirements at the expense of long latency whereas Chen et al. (2024b) offloads the attention computation on low-cost GPU devices to support economical long-sequence LLM inference. These solutions try to tackle the Long-context model from the computing system level, however, it could be better solved from algorithm advancements such as LLM quantization (Liu et al., 2024) and sparsification (Tang et al., 2024).

## 2.2 EFFICIENT INFERENCE OF LLMS

Several techniques, such as speculative decoding (Leviathan et al., 2023; Miao et al., 2024), parameter sharing (Chen et al., 2024a), and quantization (Lin et al., 2024b;a), have been proposed to improve inference efficiency. Here we focus on accelerating the attention mechanism, which poses significant time costs due to its quadratic complexity with respect to sequence length, particularly in long-context scenarios. Decoder-only transformers, trained with masked self-attention where each token depends only on preceding tokens, enable the use of key-value activation caching (KV cache) to bypass redundant computations (Pope et al., 2023). However, the KV cache can grow significantly in size. For instance, during inference with the OPT-175B model (Zhang et al., 2022) using a batch size of 512, a prompt length of 512, and an output length of 32, the KV cache demands 1.2TB of memory for storage and communication just to generate a single token (Liu et al., 2024), which poses a grand challenge for system memory and bandwidth. To tackle the challenge, one path is to reduce the KV cache storage by quantization to 2-bit or even 1-bit. Existing works (Liu et al., 2024) report to achieve 2-bit quantization without accuracy loss and (Zhang et al., 2024a) quantizes the KV cache to 1-bit with minor accuracy loss. Another orthogonal path is to focus only on the critical tokens in the KV caches, also known as token sparsity, which we will review in Sec 2.3.

## 2.3 SPARSE ATTENTION

Early works such as sparse transformer (Child et al., 2019), reformer (Kitaev et al., 2020), and longformer (Beltagy et al., 2020) made efforts to reduce attention complexity through training. Recently, Jiang et al. (2023b) trained a language model to compress prompts into smaller sets of *gist* to reduce memory pressure during caching. However, such a compression strategy requires retraining the large language model and also increases the overhead during inference.

On the other hand, many works (Ribar et al., 2023; Zhang et al., 2024b; Tang et al., 2024) propose post-training sparse attention, primarily leveraging the observation of sparse attention scores, which allows focusing on important tokens without compromising performance. StreamingLLM (Xiao et al., 2023) identifies the initial and most recent tokens as critical, while Zhang et al. (2024b;a) propose to adopt the accumulated attention score as the indicator to identify the critical tokens in the KV cache. MInference (Jiang et al., 2024) introduces a dynamic sparse pattern identification algorithm for prefilling acceleration. Additionally, Quest (Tang et al., 2024) uses min and max values to assess the importance of each KV cache page, computing only the most relevant pages, while DS (Yang et al., 2024b) focuses on selecting critical KV tokens by utilizing only important channels. However, these methods incur additional computational overhead for determining importance. In contrast, our work aims to minimize the need for such additional computations by sharing critical KV cache token indices across multiple levels.

## 3 PROBLEM FORMULATION

We begin by defining the decoding self-attention process with selected critical key-value cache tokens, which we refer to as token sparsity attention. In the decoding stage, let the query matrix be denoted as $Q \in \mathbb{R}^{1 \times d}$, the key matrix as $K \in \mathbb{R}^{n \times d}$ and the value matrix as $V \in \mathbb{R}^{n \times d}$, where $n$ denotes the

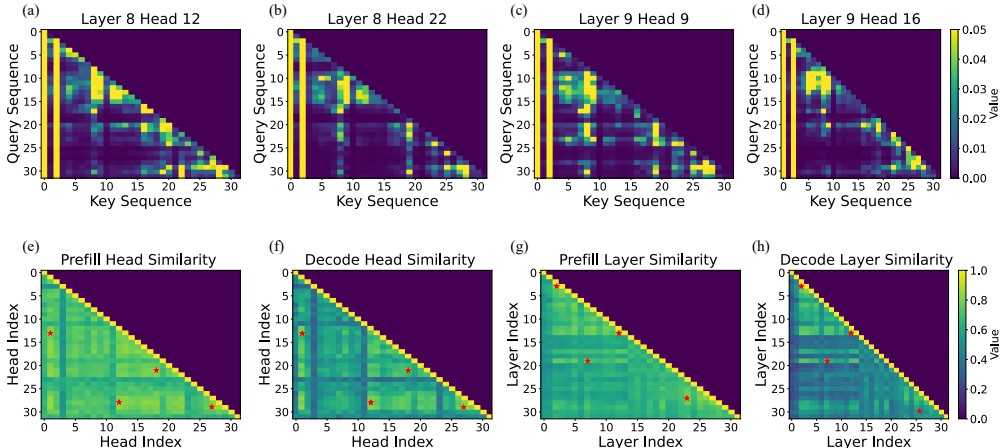

Figure 1: (a)-(d): Illustrations of self-attention matrices from a specific head in a particular layer of the LLaMA2-7b-chat model, corresponding to similar self-attention patterns observed across different layers and heads. (e)-(h) Similarity matrices of critical token indices across different heads and layers during the prefill and decode stages. The element in the $i$-th row and $j$-th column represents the similarity (Eq. 3) between the $i$-th head (layer) and the $j$-th head (layer). The red stars represent the top-k largest values in the matrix.

sequence length. Here $K_{i,*}$ represents the $i$-th row of key matrix, corresponding to the $i$-th token in the current sequence. With these notations, we define the token sparsity attention.

**Definition 3.1** (Token sparse attention, informal). *Token sparsity attention only calculates the attention weights between the query matrix and the selected critical key tokens. Let $N_c$ represent the number of KV critical tokens to be selected, and suppose we have the indices:*

$$CT = \{x_1, x_2, \ldots, x_{N_c} \mid x_i \in [0, n] \text{ and } i = 1, 2, \ldots, N_c\} \tag{1}$$

*We then select the rows $\{K_{i,*} | i \in S\}$ corresponding to these indices from the key matrix to form a new key matrix $K_{CT} \in \mathbb{R}^{N_c \times d}$, also apply the same operation to the value matrix to form a new value matrix $V_{CT} \in \mathbb{R}^{N_c \times d}$. The token sparsity ratio is defined as $\frac{N_c}{n}$ and the token sparse attention is computed through the formula shown below:*

$$y = \text{softmax}\left(\frac{Q \cdot K_{CT}^T}{\sqrt{d_h}}\right) \cdot V_{CT} \tag{2}$$

Normally, each attention block independently evaluates and selects its critical KV cache tokens, resulting in a corresponding set of indices, denoted as $CT$. We use the number of overlapping elements between two different sets to evaluate the similarity between them. Then the similarity between two critical KV cache token indices $CT_A$ and $CT_B$ can be formally written as:

$$sim(CT_A, CT_B) = \frac{|CT_A \cap CT_B|}{max(|CT_A|, |CT_B|)}, \tag{3}$$

here $|\cdot|$ represents the cardinality of a set and normally $|CT_A| = |CT_B| = N_c$. Next, we define Key-Value sharing.

**Definition 3.2** (Key-Value sharing). *Assuming that the critical KV cache token indices for attention block A are denoted as $CT_A$, if another attention block B directly reuses the indices from block A, represented as $CT_B \leftarrow CT_A$, we refer this to as key-value sharing between blocks B and A. In this case, $CT_A$ is utilized by attention block B to construct $K_{CT}$ and $V_{CT}$.*

## 4 METHOD

In this section, we first present the motivation of HShare, and then we introduce HShare in detail.

### 4.1 MOTIVATION

Previous works have pointed out not all KV tokens hold equal significance, with a limited subset, known as critical tokens, contributing most to the attention output, and these critical tokens are found to be query-aware. In this paper, we further present three observations, with insights shown in Fig. 1.

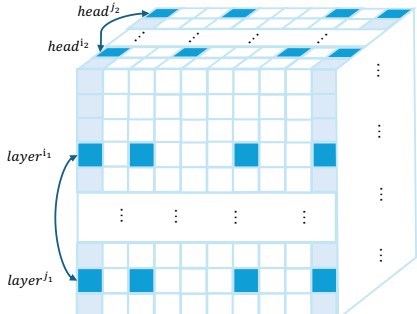

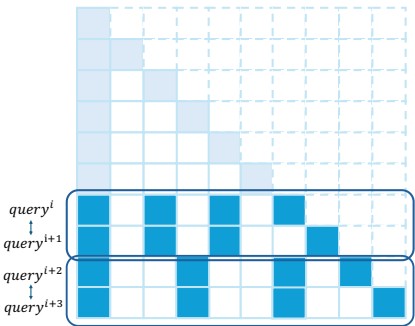

Figure 2: Share the same critical token indices between layers and heads.

Figure 3: Share the same critical token indices between adjacent queries.

**Similar critical tokens between adjacent queries**: From Fig. 1(a)-(d), we observe that within individual attention heads, the distribution of critical KV cache tokens (those associated with larger values in the attention matrix) gradually shifts with changes in the queries. However, the critical tokens across adjacent queries remain largely consistent. This observation aligns with intuition, as minor variations in query length do not significantly alter the overall sequence length, suggesting that the attention distribution remains stable. The similarity between adjacent queries presents an opportunity to reduce computational overhead by reusing the same critical KV cache tokens for every $k$ consecutive query.

**Similar sparse-patterns across different heads and layers**: The distribution of critical tokens results in different sparse patterns for attention heads. However, as illustrated in Fig. 1(a)-(d), we can also see that some sparse patterns are consistently observed across various attention heads. For example, the sparse pattern of head 12 in layer 8 is very similar to that of head 22 in the same layer. Additionally, head 9 and head 16 in layer 9 also share a similar sparse pattern, indicating that the similarity between heads exists not only within the same layer but also across different layers. This uniformity suggests that a sparse pattern derived from one head can be effectively applied to others, enabling a shared strategy for selecting critical KV cache tokens. This approach can further improve the speed and efficiency of the attention mechanism.

**Consistency relationships between prefill and decoding phases**: Fig. 1(e) and Fig. 1(f) present the head similarity matrices of layer 8 in terms of prefill stage and decoding stage, respectively. Fig. 1(g) and Fig. 1(h) present the layer similarity matrices of the whole network in terms of the prefill stage and decoding stage, respectively. We observe that the similarity matrix between prefill and decoding phases exhibits a remarkable consistency for both head-level and layer-level. This continuity underscores the potential that applying the hierarchical sharing strategy computed from the prefill phase to the decoding phase.

Motivated by the above observations, we propose HShare, a hierarchical sharing framework that shares the same critical KV cache tokens across queries, heads, and layers. HShare not only takes advantage of dynamic sparsity but also significantly reduces the computational overhead associated with critical tokens through an effective sharing mechanism.

### 4.2 HIERARCHICAL KEY-VALUE SHARING FRAMEWORK

Inspired by recent works (Zhang et al., 2024b; Xiao et al., 2023; Yang et al., 2024b), in this paper, we select critical tokens from three perspectives: initial tokens (also referred to as sink tokens), the most recent window, and significant tokens in the middle. The dark blue positions in Fig.2 and Fig.3 represent the indices of these critical tokens. Given that the decoding stage is autoregressive, each decoding step requires passing through all $L$ layers of the attention blocks. In the case of multi-head attention and group query attention, each attention block involves multiple parallel attention head $H$ computations. Consequently, to generate $n$ tokens, a total of $L \times H \times n$ attention computations are necessary, resulting in $L \times H \times n$ re-selections of critical tokens.

To reduce the overall time spent on evaluating and selecting critical tokens and further accelerate the decoding, we introduce a hierarchical key-value sharing framework. This framework is designed to facilitate key-value sharing from three perspectives: across layers, heads, and queries. Specifically, denote the attention computation of head $h$ in layer $l$ of the query $q$ as $A_{l,h,q}$, then the key-value sharing between layers $l_1$ and $l_2$ can be expressed as: $CT_{A_{l_2,h,q}} \leftarrow CT_{A_{l_1,h,q}}$.

Similarly, the key-value sharing between heads $h_1, h_2$ and queries $q_1, q_2$ can be expressed as: $CT_{A_{l,h_1,q}} \leftarrow CT_{A_{l,h_2,q}}, \quad CT_{A_{l,h,q_1}} \leftarrow CT_{A_{l,h,q_2}}$.

It should be noted that since there is a temporal dependency between the attention computations of layers and queries, it is necessary to satisfy $l_1 < l_2$ and $q_1 < q_2$ here.

Similar to the token sparsity ratio, we define the sharing ratio as a value less than or equal to 1, where a smaller value indicates a higher degree of sharing and greater computational savings. Let the total number of layers and heads be denoted as $n_l$ and $n_h$, respectively. Suppose $k_l$ layers and $k_h$ heads share critical KV token indices with other layers or heads. The sharing ratios for layers and heads are then defined as $ratio_l = 1 - \frac{k_l}{n_l}$ and $ratio_h = 1 - \frac{k_h}{n_h}$, respectively. For query-level sharing, let the total number of decoding tokens (queries) be $n_q$, and suppose $k_q$ queries bypass the selection of critical token indices by sharing key-value indices with other queries. The query-level sharing ratio is defined as $ratio_q = 1 - \frac{k_q}{n_q}$. Since the sharing across layers, heads, and queries is mutually independent, the total sharing ratio is given by:

$$sharing\ ratio = (1 - \frac{k_l}{n_l}) \times (1 - \frac{k_h}{n_h}) \times (1 - \frac{k_q}{n_q}) \tag{4}$$

### 4.3 Algorithm for Sharing configuration design

---
**Algorithm 1** Layer(Head) Sharing Algorithm

---
1: **Input:** layer(head) number $n_l(n_h)$, critical token indices matrix List $L = \{M_1, M_2, ...M_{n_l(n_h)}\}$, share number $k_l(k_h)$
2: Initialize similarity matrix $S \leftarrow zeros(n, n)$
3: **for** $i = 1$ to $n_l(n_h)$ **do**
4:     $set_i \leftarrow set(M_i.\text{flatten}())$
5:     **for** $j = 1$ to $i - 1$ **do**
6:         $set_j \leftarrow set(M_j.\text{flatten}())$
7:         overlap $\leftarrow |set_i \cap set_j|$
8:         $S_{ij} \leftarrow$ overlap / $M_i.\text{numel}()$
9:     **end for**
10: **end for**
11: **for** $i = 1$ to $n_l(n_h)$ **do**
12:     Initialize sharing config $C[i] \leftarrow i$
13: **end for**
14: **for** $i = 1$ to $k_l(k_h)$ **do**
15:     row_index, column_index $\leftarrow argmax(S)$
16:     $S[\text{row\_index}, :] \leftarrow 0$
17:     $S[:, \text{row\_index}] \leftarrow 0$
18:     $C[\text{row\_index}] \leftarrow$ column_index
19: **end for**
20: **Output:** Sharing config $C$ of layer(head).

---

For layer-level and head-level sharing, as demonstrated in Sec. 4.1, the similarity of the critical KV cache token indices shows consistency across the different queries. Therefore, we can design a sharing configuration by calculating the similarity of the critical token indices between layers and heads after prefilling, and then apply this configuration to the entire decoding phase.

Here, we use the layer-level as an example for a detailed explanation. After each prefill step, we first obtain the critical KV cache token indices matrix list for all layers $L = M_1, M_2, \ldots, M_{n_l}$, where $n_l$ represents the number of layers. Next, we compute the pairwise similarities between the critical KV cache token indices of all layers, producing a similarity matrix $S_l$. Based on the desired sharing ratio, we then perform key-value sharing between the most similar layers. Specifically, if the layer-level sharing ratio is $\beta$, meaning that $k_l = (1 - \beta) \times n_l$ layers should share critical KV token indices, we iteratively select the pair $(i, j)$ with the highest similarity (where $j < i$), allowing the $i$-th layer to reuse the indices of the $j$-th layer. Since layer $i$ now reuses the critical token indices of another layer and does not compute its own, we remove the $i$-th row and $i$-th column from the similarity matrix.

Similarly, for all heads within each layer, we apply the same method to determine the sharing configuration. Note that each layer has its own head-level sharing configuration. Algo. 1 outlines the process of calculating the similarity matrix and designing the layer (or head) sharing configuration. Formally, the output sharing configuration $C$ represents:

$$c_i = \begin{cases} j, & \text{if the } i\text{-th layer (or the } i\text{-th head) shares with the } j\text{-th layer (or the } j\text{-th head)}, \\ i, & \text{if the } i\text{-th layer needs to compute critical tokens separately.} \end{cases} \tag{5}$$

Since the sharing configuration can differ across samples, we compute it dynamically for each batch of samples.

Table 2: **Evaluation of different methods on GSM8K and COQA** and the best result (exclude origin) in each column is highlighted in bold. Complexity* refers to the theoretical time complexity for each method to select critical KV cache tokens, where $\mathcal{O}(1)$ denotes constant time complexity, and $T$ represents the theoretical computation time for a dense attention mechanism.

| Model | Architecture | Method | GSM8K(flexible/strict)↑ | COQA ↑ | Complexity* ↓ |
|---|---|---|---|---|---|
| LLaMA2-7b-chat | MHA | Original | 0.2297/0.2297 | 0.5997 | - |
| | | StreamingLLM | 0.0591/0.0023 | 0.3577 | 0 |
| | | H2O | 0.0986/0.0144 | 0.4952 | $\mathcal{O}(1)$ |
| | | Quest | 0.0478/0.0462 | 0.5713 | $0.125T$ |
| | | DS | 0.1713/0.1706 | 0.5978 | $0.0625T$ |
| | | **Ours (7/8-3/4-1/2)** | **0.1835/0.1721** | **0.5982** | $0.021T$ |
| | | **Ours (3/4-3/4-1/2)** | 0.1803/0.1703 | 0.5898 | $0.018T$ |
| | | **Ours (1/2-1/2-1/2)** | 0.1524/0.1145 | 0.5672 | $0.008T$ |
| LLaMA3-70b | GQA | Original | 0.8067/0.8052 | 0.7085 | - |
| | | StreamingLLM | 0.3912/0.0535 | 0.5460 | 0 |
| | | H2O | 0.5815/0.5588 | 0.6310 | $\mathcal{O}(1)$ |
| | | Quest | 0.4602/0.4503 | 0.6955 | $0.125T$ |
| | | DS | 0.7415/0.7392 | 0.7045 | $0.0625T$ |
| | | **Ours (7/8-3/4-1/2)** | **0.7536/0.7468** | **0.7105** | $0.021T$ |
| | | **Ours (3/4-3/4-1/2)** | 0.7512/0.7437 | 0.7010 | $0.018T$ |
| | | **Ours (1/2-1/2-1/2)** | 0.7415/0.7346 | 0.6960 | $0.008T$ |
| Mistral-7b | GQA | Original | 0.3821/0.3813 | 0.6758 | - |
| | | StreamingLLM | 0.1281/0.0085 | 0.4818 | 0 |
| | | H2O | 0.1857/0.1766 | 0.5592 | $\mathcal{O}(1)$ |
| | | Quest | 0.1342/0.1319 | 0.6378 | $0.125T$ |
| | | DS | 0.3262/0.3227 | **0.6693** | $0.0625T$ |
| | | **Ours (7/8-3/4-1/2)** | **0.3287/0.3233** | 0.6667 | $0.021T$ |
| | | **Ours (3/4-3/4-1/2)** | 0.3199/0.3154 | 0.6560 | $0.018T$ |
| | | **Ours (1/2-1/2-1/2)** | 0.2818/0.2684 | 0.6387 | $0.008T$ |

**Query-level.** As discussed in Sec. 4.1, we observe that attention weights between adjacent query tokens tend to exhibit significantly higher similarity compared to those between distant query tokens. Specifically, for query $i$ and query $i + 1$, the importance rankings of the tokens from token 0 to token $i - 1$ are largely similar. Based on this observation, we group every $a$ adjacent query to share a single critical token set, resulting in a sharing ratio of $\frac{1}{a}$ between queries. Fig.3 illustrates the sharing mechanism between two adjacent queries.

## 5 EXPERIMENT

### 5.1 ACCURACY EVALUATION

#### 5.1.1 SETUP

We evaluate HShare on GSM8K (Cobbe et al., 2021), COQA (Reddy et al., 2019), and sixteen English datasets in LongBench (Bai et al., 2023). For our evaluation, we select three widely-used models: LLaMA2-7b-chat (Touvron et al., 2023), LLaMA3-70b (Gliwa et al., 2019), and Mistral-7b (Jiang et al., 2023a), representing a range of architectures from *multi-head attention* (MHA) to *group query attention* (GQA). As baselines, we include four state-of-the-art methods: two KV cache eviction algorithms **StreamingLLM** (Xiao et al., 2023), **H2O** (Zhang et al., 2024b) and two query-aware token sparsity algorithms **Quest** (Tang et al., 2024), **DS** (Yang et al., 2024b). To ensure a fair comparison, in line with Xiao et al. (2023); Yang et al. (2024b), we keep dense attention to the prefill stage and apply token-sparse attention to all layers in the decode stage. Here, we also include the vanilla attention as **Original**, which serves as the baseline representing lossless results, without applying any sparsity in either the prefill stage or the decoding stage.

#### 5.1.2 RESULTS ON GSM8K, COQA

**Datasets and Metrics.** We use the LM-Eval framework to conduct zero-shot inference on GSM8K and COQA. GSM8K consists of around 8,000 elementary school math problems and COQA is designed for evaluating dialogue-based question-answering systems. The average context length for these two datasets is approximately 500 and 2k respectively. Here we report the GSM8K with flexible exact-match accuracy and strict exact-match accuracy, while COQA with em score.

**Inference Details.** For a fair comparison, all methods select $N_c = 128$ critical KV cache tokens and do token sparse attention, with a token sparsity ratio of approximately $1/4$ and $1/16$ for the two datasets respectively (since GSM8K consists of math problems, we select a higher sparsity

Table 3: **Evaluation of different methods on sixteen English datasets in Longbench** and the best result in each row (excluding the original) is highlighted in bold.

| Method / Dataset | Original | StreamingLLM | H2O | Quest | DS | Ours (7/8-3/4-1/2) | Ours (3/4-3/4-1/2) | Ours (1/2-1/2-1/2) |
|---|---|---|---|---|---|---|---|---|
| MultiNews | 26.22 | 23.34 | 24.88 | **26.49** | 26.30 | 26.22 | 26.05 | 25.22 |
| Musique | 8.65 | 5.79 | 6.24 | 4.62 | **8.72** | 7.52 | 8.42 | 8.09 |
| HotpotQA | 27.72 | 23.89 | 27.58 | 21.94 | 27.44 | 27.29 | 27.32 | **29.13** |
| Qasper | 21.88 | 13.80 | 17.32 | 16.96 | **20.98** | 19.98 | 20.12 | 19.60 |
| 2WikiMQA | 31.18 | 29.79 | 31.38 | 29.42 | **32.01** | 31.74 | 31.46 | 31.43 |
| Repobench-P | 52.14 | 48.06 | **51.81** | 50.24 | 48.44 | 51.33 | 50.79 | 51.63 |
| TriviaQA | 83.09 | 80.82 | 81.90 | 82.42 | 83.23 | **83.91** | **83.91** | 82.16 |
| Trec | 64.5 | 54.5 | 62.5 | **64.0** | 61.0 | 60.5 | 60.5 | 58.5 |
| Qmsum | 20.91 | 19.09 | 20.74 | **21.36** | 20.75 | 20.77 | 20.59 | 20.74 |
| NarrativeQA | 18.83 | 13.62 | 17.01 | **18.21** | 17.54 | 17.15 | 17.31 | 16.66 |
| GovReport | 26.55 | 20.64 | 23.44 | 25.46 | **27.04** | 26.63 | 25.85 | 23.93 |
| LCC | 58.26 | 55.18 | 56.55 | **56.74** | 54.87 | 55.18 | 56.23 | 56.11 |
| Passage-Count | 2.77 | 2.02 | 1.87 | 2.33 | 2.33 | 2.98 | **3.15** | 2.30 |
| Samsum | 41.01 | 37.87 | 40.18 | 40.91 | **41.30** | 40.66 | 39.77 | 40.02 |
| Passage-Retrieval-EN | 6.5 | 4.0 | 5.0 | 6.5 | 6.0 | **6.5** | 6.0 | 6.0 |
| MultifieldQA-EN | 36.15 | 23.69 | 33.64 | 31.99 | 35.83 | **36.65** | 34.08 | 32.37 |
| Average | 32.90 | 28.51 | 31.38 | 31.35 | 32.11 | **32.13** | 31.97 | 31.49 |

level, whereas COQA involves story-based question-answering, so we choose a lower sparsity level.) For HShare, we select critical tokens from three aspects, with $x = 8$ sink tokens, $y = 32$ recent tokens, and $z = 88$ critical tokens in the middle. Following the approach in Yang et al. (2024b), we load the heavy channel of the KV cache in 4-bit precision to compute the approximate attention weights, then sort them to select the top-$z$ tokens in the middle. We evaluate the effectiveness of our proposed HShare using LLaMA2-7b-chat, LLaMA3-70b, and Mistral-7b, selecting three sharing ratios **7/8-3/4-1/2**, **3/4-3/4-1/2** and **1/2-1/2-1/2** of HShare for comparison against other methods. Here **a-b-c** means HShare with layer-sharing ratio $a$, head-sharing ratio $b$, and query-sharing ratio $c$.

**Results.** The results on GSM8K and COQA, shown in Tab. 2, indicate that HShare outperforms other KV cache eviction methods (StreamingLLM and H2O). In addition, when compared to dynamic critical token selection approaches (Quest and DS), HShare can also achieve competitive performance with low complexity in selecting critical KV cache tokens. For instance, under the configuration of Ours(7/8-3/4-1/2), the accuracy of HShare is slightly superior to those in Quest and DS, but the computational complexity of HShare is only 16.8% and 33.6% of theirs, respectively. We also conduct experiments with token-sparse attention applied to both the prefill and decoding stages, the corresponding results can be found in Appendix A.1.

### 5.1.3 RESULTS ON LONGBENCH

**Datasets and Metrics.** We further use sixteen English datasets in LongBench to evaluate the performance of the proposed HShare across multiple long context benchmarks, including code completion tasks, few-shot learning tasks, document QA tasks, summarization tasks, and synthetic tasks. The metric used for each dataset can be found in Appendix A.2.

**Inference Details.** All methods select $N_c = 512$ critical KV cache tokens, with a token sparsity ratio of approximately $1/8$. Similarly, we select critical tokens from three aspects: $x = 16$ sink tokens, $y = 64$ recent tokens, and $z = 432$ critical tokens in the middle. We conduct experiments on LLaMA2-7b-chat with three sharing ratios of HShare: **7/8-3/4-1/2**, **3/4-3/4-1/2** and **1/2-1/2-1/2**.

**Results.** The results of sixteen English datasets in Longbench are shown in Tab. 3. We can find that HShare outperforms StreamingLLM, H2O, and Quest by a notable margin, which demonstrates that HShare has the ability to maintain accuracy in long-context scenarios. Additionally, the results show that HShare could achieve a competitive performance compared with DS under different sharing ratios. It is important to note that the sharing ratio of HShare serves as a hyperparameter that can influence the balance between effectiveness and efficiency. To further validate the performance of HShare, we also conduct experiments with token-sparse attention applied to both the prefill and decoding stages. More detailed results and discussion can be found in Appendix A.3.

### 5.2 EFFICIENCY EVALUATION

#### 5.2.1 SETUP

All experiments are conducted on a machine with Xeon(R) Platinum 8336C CPU, one A100 GPU, and 128G RAM.

Table 4: Attention operator latency (ms ↓) of different methods across various batch sizes and sequence lengths.

| BS | Seqlen | Flash | StreamingLLM | H2O | Quest | DS | Ours (7/8-3/4-1/2) | Ours (3/4-3/4-1/2) | Ours (1/2-1/2-1/2) |
|----|--------|-------|--------------|------|-------|-------|---------------------|---------------------|---------------------|
| 8  | 1k     | 0.230 | **0.030**    | 0.088| 0.200 | 0.141 | 0.140               | 0.124               | 0.090               |
|    | 2k     | 0.830 | **0.038**    | 0.093| 0.460 | 0.241 | 0.191               | 0.160               | 0.120               |
|    | 4k     | 1.630 | **0.420**    | 0.470| 0.850 | 0.733 | 0.602               | 0.570               | 0.530               |
| 16 | 1k     | 0.440 | **0.030**    | 0.089| 0.280 | 0.233 | 0.161               | 0.120               | 0.093               |
|    | 2k     | 1.630 | **0.073**    | 0.110| 0.770 | 0.422 | 0.270               | 0.230               | 0.190               |
|    | 4k     | 3.230 | **0.800**    | 0.850| 2.21  | 1.350 | 1.080               | 1.041               | 0.990               |

Table 5: Throughput (↑) of different methods across various batch sizes and sequence lengths.

| BS | Seqlen | GPT-Fast | StreamingLLM | H2O | Quest | DS | Ours (7/8-3/4-1/2) | Ours (3/4-3/4-1/2) | Ours (1/2-1/2-1/2) |
|----|--------|----------|--------------|-----|-------|-----|---------------------|---------------------|---------------------|
| 8  | 1k     | 228      | **264**      | 240 | 228   | 228 | 230                 | 231                 | 235                 |
|    | 2k     | 188      | **252**      | 234 | 206   | 213 | 220                 | 222                 | 226                 |
|    | 4k     | 118      | **243**      | 228 | 152   | 201 | 212                 | 214                 | 217                 |
| 16 | 1k     | 374      | **465**      | 441 | 410   | 423 | 428                 | 430                 | 439                 |
|    | 2k     | 233      | **452**      | 416 | 287   | 360 | 393                 | 398                 | 411                 |
|    | 4k     | 136      | **422**      | 396 | 175   | 286 | 336                 | 350                 | 365                 |

### 5.2.2 SELF-ATTENTION OPERATOR SPEEDUP AND END-TO-END INFERENCE SPEEDUP

**Inference Details.** Following Yang et al. (2024b), we utilized PyTorch to approximate attention, selecting the top-$z$ tokens from the middle and further incorporating $x$ initial indices and $y$ recent indices to form the complete set of critical token indices. The kernel for head-level sharing and the attention over critical KV tokens are designed using OpenAI Triton. Meanwhile, since HShare designs key value sharing between different layers and queries, we require an additional cache to store the critical token indices that need to be shared. Then for attention modules where the computation of critical token indices is bypassed due to sharing, we directly load the corresponding part from the cache. The additional storage required here is minimal, as we only need to store integers with a complexity of $\mathcal{O}(N_c)$. As for end-to-end testing, our implementation is based on GPT-fast (PyTorch, 2023), with the full attention module being replaced by our token sparsity self-attention module. We conduct the self-attention latency evaluation on a single A100 GPU with batch sizes 8, and 16 and sequence lengths ranging from 1k to 4k, with a token sparsity ratio of 1/8.

**Baseline.** In line with accuracy evaluation, here we also use two KV cache eviction algorithms **StreamingLLM**, **H2O** and two query-aware token sparsity algorithms **Quest**, **DS** as our baselines. In addition, for self-attention operator speedup evaluation, we add FlashAttention2 (**Flash**) (Dao, 2023) as our baseline, which is an optimized dense attention mechanism designed to improve the speed and efficiency and ranks among the fastest attention mechanisms. For end-to-end inference speedup evaluation, we further take **GPT-fast** (PyTorch, 2023) as the baseline for dense attention, which is acknowledged as the SOTA implementation for LLaMA models on the A100 GPU.

**Results.** Tab. 4 and Tab. 5 present the latency of attention operator and the throughput of end-to-end inference, respectively. For attention latency, we simulate the hierarchical sharing framework and average the latency over 1000 times self-attention computations. For end-to-end throughput, all results are averaged over 10 runs. Compared to dense attention, we can find that HShare consistently outperforms FlashAttention2 and GPT-fast, achieving up to an 8.6x acceleration in self-attention latency and up to a 2.7x increase in throughput. When we compare different token sparse attention methods, it can be observed that KV cache eviction algorithms have a clear advantage in system efficiency compared to query-aware token sparsity algorithms. Specifi-

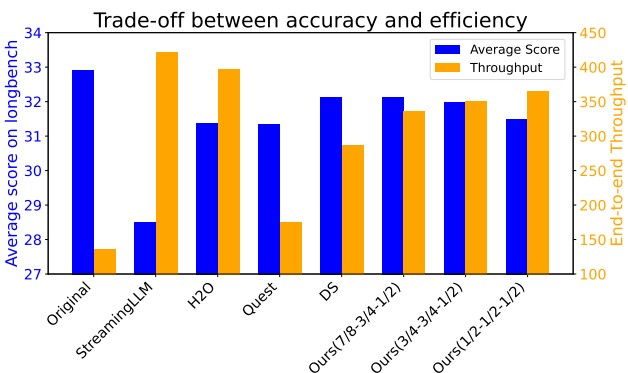

Figure 4: The average score on Longbench and the end-to-end throughput (BS=16, Seqlen=4k) of different methods

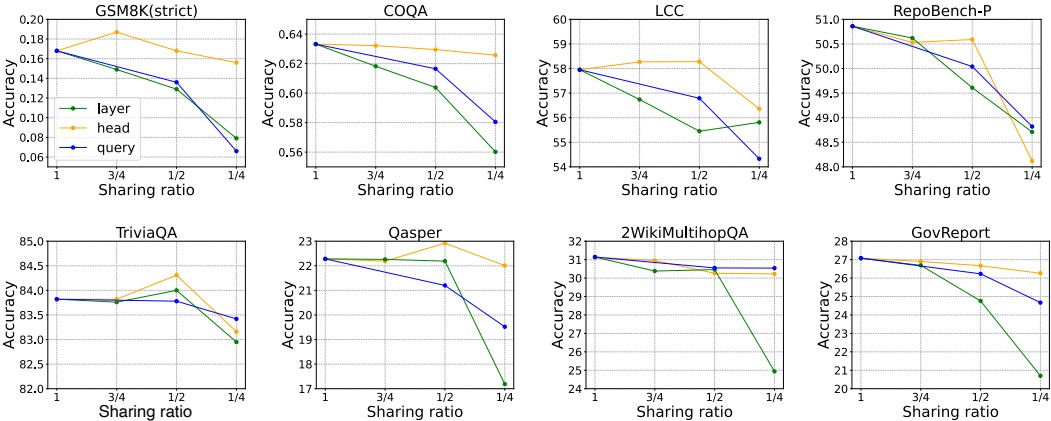

Figure 5: Results of HShare with different sharing ratios on eight datasets.

cally, StreamingLLM achieves the fastest speed as it applies a fixed sparsity pattern. However, both StreamingLLM and H2O fall short in terms of accuracy. In contrast, HShare could balance the trade-off between effectiveness and efficiency compared with other methods. For example, under the configuration of (7/8-3/4-1/2), HShare not only achieves competitive accuracy results over DS but also gains over a 9.17% improvement in end-to-end throughput when the batch size is 16 and the sequence length is 2k. Notably, as the sequence length increases, the advantage of HShare will be further amplified. For example, when the batch size is 16 and the sequence length is 4k, HShare is 17.48% faster than DS.

To further illustrate the trade-off between accuracy and efficiency, we present the average score on Longbench and the end-to-end throughput (BS=16, Seqlen=4k) of different methods in LLaMA2-7b-chat. The results are depicted in Fig. 4. We can find from Fig. 4 that the original model with the dense attention mechanism achieves the highest score over Longbench, albeit at the cost of the lowest throughput. On the other hand, StreamingLLM achieves the highest throughput, significantly outpacing other methods in processing efficiency but exhibits the lowest average score. Notably, our method performs a balance between these two aspects, showing negligible performance degradation while maintaining relatively high throughput. This demonstrates the effectiveness of our approach in achieving competitive accuracy while enhancing processing efficiency compared to other methods.

## 5.3 ABLATION STUDY

**HShare with different sharing ratios.** We conduct ablation studies under different sharing ratios on eight datasets and the results are shown in Fig. 5. Specifically, for layer-level and head-level sharing, we selected sharing ratios of $3/4$, $1/2$, and $1/4$. For query-level sharing, we group queries in sets of 2 and 4, corresponding to sharing ratios of $1/2$ and $1/4$, respectively. The results indicate that for all three levels, a smaller sharing ratio tends to result in greater accuracy loss on average. However, on certain datasets like TriviaQA and GSM8K, sharing a subset of heads leads to improved performance. Additionally, compared to head-level sharing, we observe that layer-level and query-level sharing have a greater negative impact on accuracy. Overall, the sharing ratios **3/4-3/4-1/2** and **1/2-1/2-1/2** emerge as more reasonable options. More ablation studies can be found in Appendix A.4.

## 6 CONCLUSION AND DISCUSSION

In this paper, we first systematically analyze and reveal the similarity of critical KV cache tokens across layers, heads, and query levels. Then we introduce HShare, a hierarchical framework for sharing critical KV cache token indices at all three levels to reduce the overhead associated with selecting critical KV tokens. Additionally, we propose a greedy selection algorithm for efficient sharing at the layer and head levels. Extensive evaluations show that HShare preserves model accuracy with different sharing ratios while achieving up to $8.6\times$ speedup in self-attention operations and a $2.7\times$ increase in end-to-end throughput compared with FlashAttention2 and GPT-fast respectively. To the best of our knowledge, HShare is the first work to introduce the concept of sharing critical KV cache token indices. At present, the selection of the sharing ratio remains relatively heuristic and we leave the exploration of dynamically determining whether to share and how to select an appropriate sharing ratio for different tasks as part of our future work.

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

# A    MORE EXPERIMENTS

## A.1    MORE RESULTS ON GS8K, COQA

Table 6: **Evaluation of different methods on GSM8K and COQA with sparse attention being applied to both prefill and decode stages.** The best result (exclude origin) in each column is highlighted in bold. Complexity* refers to the theoretical time complexity for each method to select critical KV cache tokens, where $\mathcal{O}(1)$ denotes constant time complexity, and $T$ represents the theoretical computation time for a dense attention mechanism.

| Model | Architecture | Method | GSM8K(flexible/strict)↑ | COQA ↑ | Complexity* ↓ |
|---|---|---|---|---|---|
| LLaMA2-7b-chat | MHA | Original | 0.2297/0.2297 | 0.5997 | - |
| | | StreamingLLM | 0.0485/0.0000 | 0.2515 | 0 |
| | | H2O | 0.0558/0.0108 | 0.3615 | $\mathcal{O}(1)$ |
| | | Quest | 0.0371/0.0364 | 0.5513 | $0.125T$ |
| | | DS | **0.1630/0.1622** | **0.6270** | $0.0625T$ |
| | | **Ours (7/4-3/4-1/2)** | 0.1580/0.1482 | 0.6093 | $0.021T$ |
| | | **Ours (3/4-3/4-1/2)** | 0.1554/0.1456 | 0.6013 | $0.018T$ |
| | | **Ours (1/2-1/2-1/2)** | 0.1319/0.0743 | 0.5960 | $0.008T$ |
| LLaMA3-70b | GQA | Original | 0.8067/0.8052 | 0.7085 | - |
| | | StreamingLLM | 0.3897/0.0311 | 0.3473 | 0 |
| | | H2O | 0.4329/0.3288 | 0.6100 | $\mathcal{O}(1)$ |
| | | Quest | 0.4708/0.4602 | 0.6900 | $0.125T$ |
| | | DS | 0.7233/0.7195 | 0.7012 | $0.0625T$ |
| | | **Ours (7/8-3/4-1/2)** | **0.7504/0.7489** | **0.7095** | $0.018T$ |
| | | **Ours (3/4-3/4-1/2)** | 0.7460/0.7445 | 0.7075 | $0.018T$ |
| | | **Ours (1/2-1/2-1/2)** | 0.7301/0.7225 | 0.6912 | $0.008T$ |
| Mistral-7b | GQA | Original | 0.3821/0.3813 | 0.6758 | - |
| | | StreamingLLM | 0.0849/0.0068 | 0.2905 | 0 |
| | | H2O | 0.0902/0.0159 | 0.4225 | $\mathcal{O}(1)$ |
| | | Quest | 0.1302/0.0569 | 0.6170 | $0.125T$ |
| | | DS | 0.3093/0.3063 | **0.6687** | $0.0625T$ |
| | | **Ours (7/8-3/4-1/2)** | **0.3145/0.3082** | 0.6631 | $0.018T$ |
| | | **Ours (3/4-3/4-1/2)** | 0.3101/0.3010 | 0.6538 | $0.018T$ |
| | | **Ours (1/2-1/2-1/2)** | 0.2775/0.2707 | 0.6313 | $0.008T$ |

Table 7: **Evaluation of different methods on sixteen English datasets in Longbench with sparse attention being applied to both prefill and decode stages.** The best result in each row (excluding the original) is highlighted in bold.

| Method / Dataset | Original | StreamingLLM | H2O | Quest | DS | Ours (7/8-3/4-1/2) | Ours (3/4-3/4-1/2) | Ours (1/2-1/2-1/2) |
|---|---|---|---|---|---|---|---|---|
| MultiNews | 26.22 | 22.93 | 22.24 | 25.74 | 25.98 | **26.30** | 25.86 | 24.83 |
| Musique | 8.65 | 4.56 | 6.32 | 6.65 | 7.16 | **8.15** | 7.63 | 7.41 |
| HotpotQA | 27.72 | 20.77 | 25.62 | 23.30 | 24.83 | 25.13 | 24.98 | **25.81** |
| Qasper | 21.88 | 14.01 | 14.13 | 17.14 | 21.94 | 21.14 | **22.13** | 21.39 |
| 2WikiMQA | 31.18 | 27.07 | 25.99 | 28.17 | 28.77 | 29.70 | **30.67** | 29.04 |
| Repobench-P | 52.14 | 48.12 | 47.67 | 44.42 | 48.78 | 50.22 | **49.67** | 48.88 |
| TriviaQA | 83.09 | 53.76 | 60.81 | 81.49 | 83.46 | **84.03** | 83.92 | 83.68 |
| Trec | 64.50 | 40.50 | 44.00 | 61.00 | **61.50** | 59.00 | 59.00 | 57.50 |
| Qmsum | 20.91 | 18.81 | 19.28 | **20.80** | 20.22 | 20.40 | 20.01 | 20.28 |
| NarrativeQA | 18.83 | 9.52 | 11.83 | 14.53 | 15.76 | 15.29 | 16.05 | **16.40** |
| GovReport | 26.55 | 21.39 | 21.51 | 25.95 | **26.53** | 25.87 | 25.76 | 23.88 |
| LCC | 58.26 | 52.41 | 57.47 | 53.99 | **57.75** | 57.20 | 56.29 | 55.89 |
| Passage-Count | 2.77 | 2.54 | 2.29 | **4.77** | 2.21 | 2.27 | 2.27 | 2.50 |
| Samsum | 41.01 | 37.37 | 38.57 | 40.67 | 41.48 | **42.06** | 40.73 | 40.13 |
| Passage-Retrieval-EN | 6.5 | 3.5 | 3.5 | 3.5 | 9.0 | **9.5** | 9.0 | 6.0 |
| MultifieldQA-EN | 36.15 | 20.75 | 21.10 | 21.61 | **37.55** | 36.80 | 34.46 | 34.21 |
| Average | 32.90 | 24.86 | 26.40 | 29.61 | 32.06 | **32.07** | 31.78 | 31.11 |

We also conduct experiments with sparse attention applied to both the prefill and decoding stages. The results on GSM8K and COQA are shown in Tab. 6. Compared to Tab. 2, the performance of most methods degrades due to the introduction of sparsity during the prefill stage. In this setting, HShare still significantly outperforms other KV cache eviction methods (StreamingLLM and H2O). On LLaMA2-7b-chat, HShare also incurs minimal accuracy loss while it achieves higher accuracy on LLaMA3-70b. We attribute this to the more redundancy in LLaMA3-70b for these tasks, where HShare may serves a regularization role. Notably, we also observe a significant performance drop in H2O. We suspect this is because H2O performs topK operations based on historical attention scores

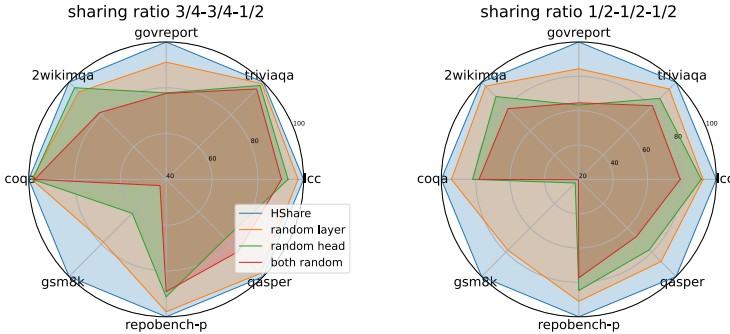

Figure 6: Comparison between HShare and random sharing at the layer level and head level. We take HShare as the 100% baseline and normalize the results of others accordingly.

to achieve sparsity. As a result, when sparsity is applied during the prefill stage, the topK operation during decoding accesses less historical information, leading to a considerable accuracy loss.

## A.2 DETAILS OF LONBENCH

We report MultiNews, Qmsum, GovReport, Samsum with rouge score; Musique, HotpotQA, Qasper, 2WikiMQA, TriviaQA, NarrativeQA, MultifieldQA-EN with F1 score; trec with classification accuracy; Passage-Count, Passage-Retrieval-EN exact-match accuracy; RepoBench-P and LCC with similarity score.

## A.3 MORE RESULTS ON LONGBENCH

Here, we also conduct experiments with sparse attention being applied to both prefill and decode stages. The results across the 16 commonly used English datasets from Longbench are shown in Tab. 7. It can be seen that HShare achieves the highest scores in the most sub-datasets. On average, HShare outperforms StreamingLLM, H2O, and Quest, and can still achieve competitive results with DS under a sharing ratio of 7/8-3/4-1/2. Compared with Tab. 3, the results show slight degradation, while H2O shows significant deterioration.

## A.4 MORE ABLATION STUDIES

**Ablation study on different sharing schemes.** We further conduct ablation studies to evaluate our layer(head) sharing scheme as shown in algorithm 1. We compare our greedy sharing scheme with random sharing scheme under the same sharing ratio and the corresponding results are shown in Fig. 6. The empirical results indicate that when critical token indices are shared by randomly selected layers and heads, the accuracy decreases significantly across all datasets, further validating the effectiveness of sharing critical tokens between similar layers and heads.

Table 8: Performance of HShare on MultiNews.

| Sharing Ratio | 1K | 2k | 3k | 4k |
|---|---|---|---|---|
| Ours-7/8-3/4-1/2 | 27.78 | 27.18 | 25.87 | 23.51 |
| Ours-3/4-3/4-1/2 | 27.77 | 27.55 | 25.73 | 22.49 |
| Ours-1/2-1/2-1/2 | 26.58 | 25.93 | 24.65 | 19.99 |

**Ablation study on different context lengths.** To test our method in document summarization with varying context lengths, we conduct experiments on the *MultiNews* dataset, which belongs to the document summarization category. We evaluate the proposed HShare across different context lengths and sharing ratios. The results are presented in Tab. 8. It should be noted that regardless of the length of the text, we consistently selected 256 critical tokens. The results show that the score decreases as the context length increases, and similarly, higher degrees of sharing also lead to a decline in performance. This suggests that when dealing with long context lengths, a more moderate sharing

strategy is needed to maintain accuracy, whereas, for shorter texts, a higher degree of sharing can be applied.

Table 9: Attention latency (ms ↓) and GSM8K accuracy of HShare under different sharing ratios.

| Sharing Ratio | Attention Latency(ms) | GSM8K(flexible/strict) |
| --- | --- | --- |
| 1/2 | 0.31 | 0.144/0.136 |
| 1/4 | 0.23 | 0.135/0.125 |
| 1/8 | 0.19 | 0.132/0.074 |
| 1/16 | 0.10 | 0.107/0.032 |

**Trade-off within HShare.** In addition, we provide the attention latency and GSM8K accuracy of our proposed HShare under different sharing ratios in Tab. 9 to reveal the tradeoff between accuracy and efficiency within HShare.

## B    DISCUSSION OF POTENTIAL ADAPTATIONS

We believe HShare can effectively support transformer variants. HShare aims to optimize long sequence attention operations by sharing critical key and value indices between layers, heads, and queries. As such, convolutional or graph neural networks with attention can also benefit from the proposed HShare. Below are two examples of potential applications:

- Convolutional neural network (CNN) with attention: (Son et al., 2024) proposes to adopt CNNs to extract image features and use spatial-temporal attention to identify crucial frames. To apply HShare in this network, we can predict the indices of critical features (similar to critical tokens), which can then be shared across heads and layers. The potential adaptation involves using only the critical features for attention computation. Since video input exhibits temporal redundancy across nearby frames, critical feature indices can also be shared across frames. HShare can further be applied to other CNN-related works, such as VQA (Anderson et al., 2018).

- Graph neural network (GNN) with attention: Graph neural networks have been applied in various fields (Li et al., 2023b;c; Chen et al., 2024c; Li et al., 2024; Bao et al., 2024; Zheng et al., 2025) and Veličković et al. (2017) adopts multi-head attention to extract features from a set of input nodes. HShare can be seamlessly applied to identify critical nodes and share their indices across heads and layers (if multiple layers are used). Depending on the problem, if the graph nodes are close and similar, the key indices of these nodes can also be shared across nodes, similar to query sharing in HShare. Similar works (Thekumparampil et al., 2018; Wu et al., 2021) could also benefit from applying HShare to reduce computational load. Moreover, HShare demonstrates significant potential in graph-related tasks that leverage graph transformers, particularly those involving large-scale graphs or high computational complexity. This is especially relevant in fields such as biology, where tasks related to molecules and proteins are prevalent (Krapp et al., 2023; Wu et al., 2024; Yang et al., 2024a), as well as in quantum-related applications (Bao et al., 2025).

