# OpenReview forum: "HShare: Fast LLM Decoding by Hierarchical Key-Value Sharing"
_ICLR.cc/2025/Conference — ICLR 2025 Poster_

### Official Review · Reviewer_Bqih · 2024-10-17

**Soundness:** 4
**Presentation:** 3
**Contribution:** 4
**Rating:** 8
**Confidence:** 3

**Summary:**

This paper discusses HShare, a hierarchical framework that facilitates the sharing of critical KV cache token indices across layers, heads, and queries. From observed similarities in KV token criticality across different levels, HShare enables a more efficient sharing strategy. The paper also introduces a greedy algorithm to dynamically determine the optimal sharing configuration for the decoding phase. The experimental results demonstrate that HShare can maintain similar accuracy and improve the end-to-end throughput.

**Strengths:**

1. The problem is very practical and of high interest to the community, especially with limited computation resources
2. A very comprehensive set of experiments to showcase the performance (latency and accuracy) on different datasets.
3. Well-organized paper with adequate discussions on other existing methods.

**Weaknesses:**

1. The paper should have some figures that discuss the tradeoff between accuracy and efficiency

**Questions:**

1. What would be your suspected reason for the performance gain by enabling KV sharing in Figure 6?
2. The sharing ratio can be more finely tuned. Do you think the sharing ratios would be dependent on the dataset and the task?

---

> ### Author Response · Authors · 2024-11-20
> **Rebuttal (1/2)**
>
> Thank you for acknowledging our work. Your rating has been a great source of encouragement for us. Below is our detailed response.
>
>
> > **W1**: The paper should have some figures that discuss the tradeoff between accuracy and efficiency.
>
>
> Thank you for the valuable suggestion. Here we provide the attention latency and GSM8K accuracy of our proposed HShare under different sharing ratios in Table 1 to reveal the tradeoff between accuracy and efficiency within our method HShare. We have also included a figure to present the average score on Longbench and end-to-end throughput of various methods, highlighting the trade-off between accuracy and efficiency across all methods. Please refer to Figure 7 in the revised paper.
>
> In addition, we provide system efficiency comparisons with other baselines, including two KV cache eviction algorithms (StreamingLLM and H2O) and two query-aware token sparsity algorithms (Quest and DS). From the results, it can be observed that KV cache eviction algorithms have a clear advantage in system efficiency compared to query-aware token sparsity algorithms. Specifically, StreamingLLM achieves the fastest speed as it applies a fixed sparsity pattern. However, both StreamingLLM and H2O fall short in terms of accuracy. In contrast, HShare not only preserves accuracy but also achieves significant speedup compared to other query-aware dynamic token sparsity methods. For example, in terms of attention latency when batch size is 16 and sequence length is 2k, our method delivers up to a 2.80x speedup over Quest and a 2.21x speedup over DS. We have also added these tables into our revised manuscript (Appendix).
>
>
>
> **Tabel 1 Attention latency ($\downarrow$) and GSM8K accuracy of HShare under different sharing ratios.**
>
> | Sharing Ratio | Attention latency(ms) | GSM8K (flexible/strict)  |
> |---------------|------------|----------------|
> | 1/2           | 0.31    | 0.144/0.136  |
> | 1/4           | 0.23    | 0.135/0.125   |
> | 1/8           | 0.19    | 0.132/0.074 |
> | 1/16          | 0.10     | 0.107/0.032  |
>
>
>
> **Table 2 Attention latency ($\downarrow$) of different methods across various batch sizes and sequence lengths.**
>
>
> | bs  | seq_len | Flash (dense) | StreamingLLM | H2O  | Quest | DS   | Ours (3/4-3/4-1/2) | Ours (1/2-1/2-1/2) |
> | --- | ------- | ------------- | ------------ | ---- | ----- | ---- | ------------------ | ------------------ |
> | 8   | 1k      | 0.23          | 0.03         | 0.09 | 0.20  | 0.14 | 0.12               | 0.09               |
> | 8   | 2k      | 0.83          | 0.04         | 0.10 | 0.46  | 0.24 | 0.16               | 0.12               |
> | 8   | 4k      | 1.63          | 0.42         | 0.47 | 0.85  | 0.73 | 0.57               | 0.53               |
> | 16  | 1k      | 0.44          | 0.03         | 0.09 | 0.28  | 0.13 | 0.12               | 0.09               |
> | 16  | 2k      | 1.63          | 0.07         | 0.11 | 0.77  | 0.42 | 0.23               | 0.19               |
> | 16  | 4k      | 3.23          | 0.80         | 0.85 | 2.21  | 1.35 | 1.04               | 0.99               |
>
>
>
> **Table 3 End-to-end throughput ($\uparrow$) of different methods across various batch sizes and sequence lengths.**
>
> | bs  | seq_len | Flash (dense) | StreamingLLM | H2O | Quest | DS  | Ours (3/4-3/4-1/2) | Ours (1/2-1/2-1/2) |
> | --- | ------- | ------------- | ------ | --- | ----- | --- | ------------------ | ------------------ |
> | 8   | 1k      | 228           | 264    | 240 | 228   | 228 | 231                | 235                |
> | 8   | 2k      | 188           | 252    | 234 | 206   | 213 | 222                | 226                |
> | 8   | 4k      | 118           | 243    | 228 | 152   | 201 | 214                | 217                |
> | 16  | 1k      | 374           | 465    | 441 | 410   | 423 | 430                | 439                |
> | 16  | 2k      | 233           | 452    | 416 | 287   | 360 | 398                | 411                |
> | 16  | 4k      | 136           | 422    | 396 | 175   | 286 | 350                | 365                |
>
>
>
> > **Q1**: What would be your suspected reason for the performance gain by enabling KV sharing in Figure 6?
>
> Thank you for your question. We have also observed that a certain degree of sharing may improve performance on some datasets, especially when we only perform head-level sharing. We suspect that this may be due to the model architecture and the sharing of critical tokens plays a role in regularization. Specifically, LLaMA2-7b-chat uses multi-head attention with 32 heads, some of these heads show high similarity in certain layers. When the sharing rate is relatively small, the regularization would improve performance to a certain extent.

---

> > ### Author Response · Authors · 2024-11-20
> > **Rebuttal (2/2)**
> >
> > > **Q2**: The sharing ratio can be more finely tuned. Do you think the sharing ratios would be dependent on the dataset and the task?
> >
> >
> > Yes. We believe that different sharing ratios have varying effects across different datasets or tasks. The table below shows the performance of HShare on the GSM8K and TriviaQA datasets as the sharing ratio decreases from 1 to 1/16, with a sparsity ratio of 1/8. GSM8K is a more challenging mathematical reasoning task, while TriviaQA is a simpler question-answering task. We can find that with the degree of sharing increases, the performance on GSM8K declines noticeably. When the sharing ratio reaches 1/32, accuracy drops to about 50% of its original value, whereas on TriviaQA, the decline is almost negligible. Therefore, we intuitively believe that for simpler, text-based tasks, a higher sharing ratio can be used to achieve greater acceleration, while for more difficult, reasoning-based tasks, a more moderate sharing ratio should be applied. Currently, our approach is still a relatively empirical sharing strategy and we plan to explore a greedy method for tuning the sharing ratio as part of our future work.
> >
> > **Table 4 Performance of HShare on different tasks with different sharing ratios under a sparsity of 1/8.**
> > | sharing ratio | GSM8K  (flexible/strict)     | Triviaqa |
> > |-------|-------------|----------|
> > | 1     | 0.167/0.167 | 83.82    |
> > | 1/4   | 0.135/0.125 | 83.68    |
> > | 1/8   | 0.132/0.074 | 83.68    |
> > | 1/16  | 0.107/0.032 | 83.51    |
> > | 1/32  | 0.083/0.017 | 83.09    |
> > | 1/64  | 0.076/0.012 | 82.28    |
> >
> >
> > ---
> > We hope our detailed response could answer your questions and address your concerns, looking forward to receiving your feedback soon.

---

> > > ### Author Response · Authors · 2024-11-25
> > >
> > > Dear Reviewer Bqih,
> > >
> > > Thank you for your valuable comments on our paper. Since the discussion phase deadline nears, we look forward to your responses. If needed, we are happy to provide more responses.
> > >
> > > Best wishes,
> > >
> > > the authors

---

### Official Review · Reviewer_RqY3 · 2024-10-31

**Soundness:** 3
**Presentation:** 2
**Contribution:** 3
**Rating:** 8
**Confidence:** 3

**Summary:**

HShare aims to reduce computation complexity of key value caches of transformer-based LLMs by sharing theses caches between attention heads within a layer, the layers themselves and queries within a batch. They identify critical tokens from three areas: initial/sink tokens, most recent window and significant tokens in the middle. For each of three dimensions (head, layer, query) they identify elements in these dimensions that are similar (based on a self-defined metric: overlap of cached key-value pairs) and only compute the respective key-value cache once, which is then shared between the respective elements. The resulting reduction in computation improves the latency and throughput for most configurations while maintaining the accuracy.

**Strengths:**

* A big strength is the evaluation. There is a wide variety of datasets used with a reasonable number of models, which show some variance in model size and architecture. Four additional framework are chosen to serve as baseline for comparison. The evaluation not only focuses on accuracy, but on compute performance (latency, throughput) as well. The provided ablation study is also fair.
* The main idea is presented clearly for the most part and is reasonable motivated with examples/microbenchmarks.
* The reduction in computation complexity with a subsequent performance improvement seems like a relevant contribution.

**Weaknesses:**

* The language is a big issue. I had to read several sentences multiple times to make sense of them. So I suggest to improve the writing, so that language reads well.
* The cache hierarchy is not discussed, i.e. the order in which the three dimensions are applied, which data they might share and so on. The individual levels are discussed in how they are constructed and their details, but not how they are applied together apart from that they are independent from each other (in their memory benefits), which contradicts the initial hierarchy claim.
* some of the equations do not add much (eq. 4 and 5)
* The performance of the original models without modification is kind of missing as a baseline for the evaluation.

more specific issues:
* line 64: "Although these two methods only load critical tokens, they retain all the KV cache"
  * That seems like a contradiction.
* 2.1: recent efforts are from 2023 and 2024, but their downsides are discussed in a paper from 2022 - seems counterinituitive
* line 248: "Fig. 1(c) and Fig. 1(d)" - I think these are not the correct subfigures: it should be 1e and 1f. I think "Fig. 1(e) and Fig. 1(f)" in line 250 is similarly wrong.
* line 404: first time that the full name "DoubleSparse (DS)" was introduced; all previous mentions just refer to DS
* table 3: I question the validity of computing an average from different types of scores (similarity, F1 and rouge).
* I think the table and figure placement for the evaluation can be improved, i.e. moved closer to where they are discussed.

detailed/minor issues:
* language:
  * line 40/41: "since the context length expands" - probably better "since as the context length expands"
  * line 43: "to addressing this issue" - I think it should either be "to address this issue" or "for addressing this issue".
  * line 45: "many works manage to only load these critical KV cache tokens" - I think "manage" is the wrong word here, maybe better: "many works choose to only load these critical tokens into the KV cache"
  * line 135: "make efforts" - should probably be past tense: "made efforts"
  * line 136: "Jiang et al. (2023b) trains" - train (multiple authors as subject), or even better "trained"
  * line 186: "also the same operation for value matrix and form a new value" - missing verb and general improvement: "also apply the same operation to the value matrix to form a new value matrix ..."
  * line 265: "layers of attention blocks" - I think it should be "layers of the attention blocks"
* typos:
  * line 46: "tokens(also" - missing whitespace
  * line 85: "GSM8K Cobbe et al. (2021)" - missing brackets around the reference, probably the wrong cite command is used
  * line 120: "scenarios.Decoder-only" - missing whitespace
  * line 143: "MInference(Jiang et al., 2024)" - missing whitespace
  * line 182: "i-th head(layer) and the j-th head(layer)" - missing whitespace between "head(layer)", two times; alternatively you could also use "head/layer" or "head|layer" (also applies to other occurences of the same phrase)
  * line 186: "S}corresponding" - missing whitespace
  * line 210: "HSahre" -> "HShare"
  * line 363: "Sec.4.1" - missing whitespace
  * line 408: "We use LM-Eval framework" - missing "the" before "LM-Eval"
  * Figure 6: x-axis label for TrivialQA is "Sparsity Level", when "Sharing ratio" is used for all other subfigures
* references:
  * "Anze Xie Ying Sheng Lianmin Zheng Joseph E. Gonzalez Ion Stoica Xuezhe Ma Dacheng Li*, Rulin Shao* and Hao Zhang" - author list is broken: wrong order, missing commas between the authors and please also remove the asterisks
  * consider using title case for the titles

**Questions:**

* line 49: What are "heavy tokens"?
* line 137: "However, such a compression strategy ... and also increases the overhead during inference." - In what regard? Wasn't the
 compression of prompts applied to reduce certain types of overheads, for example memory complexity?
* Figure 2: On what query sequence is the figure based?
* 5.2.2 Where does the query data come from?

---

> ### Author Response · Authors · 2024-11-20
> **Rebuttal (1/2)**
>
> Thank you for acknowledging our work. Your rating has been a great source of encouragement for us. We truly appreciate the time and effort you've dedicated to reviewing our work, as well as your thoughtful and incredibly detailed feedback.
>
>
> > **W1**: The language is a big issue. I had to read several sentences multiple times to make sense of them. So I suggest to improve the writing, so that language reads well.
>
>
> Based on your valuable comments, we have revised the language to improve clarity and readability, with the hope that it will now be easier to follow. Your suggestions are very helpful in refining the manuscript, and we are grateful for your contribution to improving its quality.
>
> > **W2**: The cache hierarchy is not discussed, i.e. the order in which the three dimensions are applied, which data they might share and so on. The individual levels are discussed in how they are constructed and their details, but not how they are applied together apart from that they are independent from each other (in their memory benefits), which contradicts the initial hierarchy claim.
>
>
> Thank you for your suggestions. Here, we provide an illustration of the hierarchical sharing architecture (https://postimg.cc/w1v0qpnm). In details, HShare first evaluates whether the current query should share critical token indices with those in the previous query. If this condition is met, there’s no need to proceed with layer-level and head-level sharing. If not, HShare then checks whether the current layer should share critical token indices with those in its previous layer. If layer-level sharing is applied, head-level sharing will not be performed. Only when query sharing and layer sharing are not performed, the head-level sharing will be considered according to the calculated sharing config.
>
> We have reported the results of the three levels working together as well as their individual effects in our paper. However, they can also be combined in any combination. Here, we report the results of pairwise combination sharing in Table 1, with the sharing ratio for each level set to 1/2.
>
>
> **Table 1 The results of pairwise combination sharing.**
>
> | Sharing Strategy             | LCC   | Repobench-P | TriviaQA | Qasper | 2WikiMQA | GovReport | GSM8K(flexible/strict) | COQA  |
> |------------------------------|-------|-------------|----------|--------|----------|------------|-------------------------|-------|
> | Not share                   | 57.95 | 50.86       | 83.82    | 22.28  | 31.14    | 27.08      | 0.167/0.168            | 0.6282|
> | Share layer+head+query  | 55.89 | 48.88       | 83.66    | 21.39  | 29.04    | 23.88      | 0.132/0.074            | 0.5960|
> | Share query+layer  | 56.88 | 49.55       | 83.66    | 20.35  | 30.35    | 23.96      | 0.130/0.099            | 0.5980|
> | Share query+head   | 56.98 | 50.2        | 84.75    | 21.53  | 31.15    | 26.05      | 0.135/0.125            | 0.6182|
> | Share layer+head  | 55.96 | 49.86       | 84.72    | 21.52  | 28.99    | 24.06      | 0.151/0.126            | 0.6008|
>
>
>
>
>
> > **W3**: some of the equations do not add much (eq. 4 and 5)
>
> Thanks, we have revised the corresponding part in the new manuscript.
>
>
> > **W4**: The performance of the original models without modification is kind of missing as a baseline for the evaluation.
>
> Thank you for your valuable suggestion. We appreciate your insight and follow your recommendation, we have already added the performance of the unmodified models as a baseline for the evaluation in the revised manuscript. We hope this could help provide a clearer context for the results.
>
>
> **Specific Issues**:
> > line 64: "Although these two methods only load critical tokens, they retain all the KV cache" That seems like a contradiction.
>
> These two methods retain all the KV cache, but they only select critical tokens to participate in the attention output calculation. Since these methods use query-aware token sparsity, meaning that the critical tokens selected change each time, they need to retain all the KV cache instead of performing KV cache eviction.
>
> > Table 3: I question the validity of computing an average from different types of scores (similarity, F1 and rouge).
>
> Averaging the scores of different metrics doesn't have intrinsic meaning, but we do it because we believe it can somewhat reflect the overall performance of the algorithm across different datasets. And here we humbly point out that some other works[1,2] also report the average score on Longbench.
>
> [1] InfLLM: Training-Free Long-Context Extrapolation for LLMs with an Efficient Context Memory. NeurIPS, 2024.
>
> [2] Palu: Compressing KV-Cache with Low-Rank Projection. Arxiv, 2024.

---

> > ### Author Response · Authors · 2024-11-20
> > **Rebuttal (2/2)**
> >
> > > **Writing issues**:
> >
> > As for the other **specific issues, language and typos**, we have already made the necessary revisions to the manuscript. Thank you once again for the time and effort you have dedicated to reviewing our work; it is truly an honor to have a reviewer like you.
> >
> >
> > > **Q1**: line 49: What are "heavy tokens"?
> >
> > Thanks for the question. In query-aware token sparsity approaches, tokens are often classified as "critical" or "non-critical" based on their importance, "critical" tokens have a large attention weight while "non-critical" tokens always have an attention weight near zero. Critical tokens are likely to be retained for full computation, whereas non-critical tokens might be pruned.  **Here "heavy tokens" means those "critical tokens".** We apologize for the confusion caused by the terminology and we have revised the corresponding expressions in the manuscript.
> >
> > > **Q2**: line 137: "However, such a compression strategy ... and also increases the overhead during inference." - In what regard? Wasn't the compression of prompts applied to reduce certain types of overheads, for example memory complexity?
> >
> >
> > Thank you for your comment. You are correct that prompt compression helps reduce memory complexity. However, in this context, the term "overhead during inference" refers to the additional computational cost incurred due to the use of the compressed model. Specifically, instead of solely performing inference on the large model, we now must also perform inference on both the large model and the compressed model. We hope this clarifies the point.
> >
> >
> > > **Q3**: Figure 2: On what query sequence is the figure based?
> >
> > Thanks for the question. Figure 2 is an illustration of our sharing idea, which means it is not based on any exact query sequence. Apologies for any misunderstanding caused, and we have further clarified this point in the revised manuscript.
> >
> >
> > > **Q4**: 5.2.2 Where does the query data come from?
> >
> > Thanks for the question. The query data comes from the wikitext2 dataset.
> >
> > ---
> > We hope our response could answer your questions and address your concerns, looking forward to receiving your feedback soon.

---

> > > ### Comment · Reviewer_RqY3 · 2024-11-23
> > >
> > > Thank you for the answers to my questions and points raised as well as the revision of the submission.

---

> > > > ### Author Response · Authors · 2024-11-24
> > > >
> > > > Thank you for recognizing our work and for your timely response!

---

### Official Review · Reviewer_DdXB · 2024-11-01

**Soundness:** 3
**Presentation:** 3
**Contribution:** 3
**Rating:** 6
**Confidence:** 3

**Summary:**

The paper introduces HShare, a hierarchical key-value (KV) sharing framework to accelerate inference in large language models (LLMs). HShare exploits observed similarities in critical KV cache token usage across layers, heads, and adjacent queries to reduce the computational overhead of selecting query-specific tokens repeatedly. It introduces a method to share critical KV cache token indices within and across layers, heads, and queries and proposes a greedy algorithm to optimize layer- and head-level sharing configurations. Experiments demonstrate that HShare maintains accuracy while providing up to 8.6× speedup in self-attention and a 2.7× improvement in end-to-end throughput across various LLM architectures, including LLaMA and Mistral models.

**Strengths:**

The hierarchical sharing of KV cache tokens across heads, layers, and queries is an innovative approach to reducing latency in self-attention. HShare's observation of token reuse across different levels of attention blocks offers a new angle to address computational efficiency in LLM decoding. HShare's ability to maintain high accuracy while delivering substantial improvements in self-attention latency and throughput makes it significant for practical deployment scenarios. The method's compatibility with several model types suggests its value for LLM inference optimization.

**Weaknesses:**

While HShare achieves substantial speedup, the layer- and head-level sharing configurations are tested only on a few model architectures, leaving broader applicability unexplored. Additionally, its performance in tasks that demand highly dynamic attention, such as document summarization with varying context lengths, could be tested to reveal any potential trade-offs. Minor computational overhead remains from online calculation in the prefill phase, though this appears manageable.

**Questions:**

Can HShare effectively support transformer variants such as convolutional or graph neural networks? A brief discussion on potential adaptations would help generalize its usability.

---

> ### Author Response · Authors · 2024-11-20
> **Rebuttal**
>
> Thank you for acknowledging our work and your suggestions are very helpful. Below is our detailed response.
>
> > **W1**: While HShare achieves substantial speedup, the layer- and head-level sharing configurations are tested only on a few model architectures, leaving broader applicability unexplored. Additionally, its performance in tasks that demand highly dynamic attention, such as document summarization with varying context lengths, could be tested to reveal any potential trade-offs. Minor computational overhead remains from online calculation in the prefill phase, though this appears manageable.
>
> Thank you for the valuable suggestions. We have discussed potential adaptations of our method to other model architectures. For more details, please refer to the answer to **Q1**.
>
> To test our method in document summarization with varying context lengths, we conduct experiments on the MultiNews dataset, which belongs to the document summarization category. We evaluate the proposed HShare across different context lengths and sharing ratios. The results are presented in Table 1. It should be noted that regardless of the length of the text, we consistently selected 256 critical tokens. The results show that the score decreases as the context length increases, and similarly, higher degrees of sharing also lead to a decline in performance. This suggests that when dealing with long context lengths, a more moderate sharing strategy is needed to maintain accuracy, whereas for shorter texts, a higher degree of sharing can be applied. We have added the above description and experiment into the revised manuscript  (Appendix).
>
> **Table 1. Performance of HShare on MultiNews.**
>
> | Sharing Ratio (layer-head-query) | 1K    | 2K    | 3K    | 4K    |
> |----------------------------------|-------|-------|-------|-------|
> | Ours-3/4-3/4-1/2                 | 27.77 | 27.55 | 25.73 | 22.49 |
> | Ours-1/2-1/2-1/2                 | 26.58 | 25.93 | 24.65 | 19.99 |
> | Ours-1/4-1/4-1/4                 | 25.39 | 23.44 | 22.47 | 19.21 |
>
>
>
>
> > **Q1**: Can HShare effectively support transformer variants such as convolutional or graph neural networks? A brief discussion on potential adaptations would help generalize its usability.
>
>
> Yes, we believe HShare can effectively support transformer variants. HShare aims to optimize long sequence attention operations by sharing critical key and value indices between layers, heads, and queries. As such, convolutional or graph neural networks with attention can also benefit from the proposed HShare. Below are two examples of potential applications:
>
> CNN with attention: [1] proposes to adopt CNNs to extract image features and use spatial-temporal attention to identify crucial frames. To apply HShare in this network, we can predict the indices of critical features (similar to critical tokens), which can then be shared across heads and layers. The potential adaptation involves using only the critical features for attention computation. Since video input exhibits temporal redundancy across nearby frames, critical feature indices can also be shared across frames. HShare can further be applied to other CNN-related works, such as VQA [2].
>
>
> Graph neural network with attention: [3] adopts multi-head attention to extract features from a set of input nodes. HShare can be seamlessly applied to identify critical nodes and share their indices across heads and layers (if multiple layers are used). Depending on the problem, if the graph nodes are close and similar, the key indices of these nodes can also be shared across nodes, similar to query sharing in HShare. Similar works [4, 5] can also benefit from applying HShare to reduce computational load.
>
> The discussion will be added to the revised manuscript.
>
>
> **References**
>
> [1] CSTA: CNN-based Spatiotemporal Attention for Video Summarization. CVPR, 2024.
>
> [2] Bottom-up and top-down attention for image captioning and visual question answering. CVPR, 2018.
>
> [3] Graph attention networks, ICLR, 2018.
>
> [4] Attention-based graph neural network for semi-supervised learning. arXiv,2018.
>
> [5] Representing long-range context for graph neural networks with global attention. NeurIPS, 2021.
>
>
> ---
>
> We hope this response could answer your questions and address your concerns, looking forward to receiving your further feedback soon.

---

> > ### Author Response · Authors · 2024-11-25
> >
> > Dear Reviewer DdXB,
> >
> > Thank you for your valuable comments on our paper. Since the discussion phase deadline nears, we hope our responses have answered your questions and addressed your concerns. If needed, we are happy to provide more responses.
> >
> > Best wishes,
> >
> > the authors

---

> > > ### Comment · Reviewer_DdXB · 2024-12-02
> > >
> > > Thank you for your reponse.
> > >
> > > I have read through them and would like to remain my original score.
> > >
> > > Thanks.

---

> > > > ### Author Response · Authors · 2024-12-02
> > > >
> > > > Thank you for recognizing our work. We truly appreciate the time and effort you've dedicated to reviewing our work.

---

### Official Review · Reviewer_ZRXS · 2024-11-03

**Soundness:** 3
**Presentation:** 2
**Contribution:** 2
**Rating:** 6
**Confidence:** 4

**Summary:**

The authors conducted systematic experiments to analyze the similarity of KV cache selection among queries, heads, and layers. Capitalizing on these findings, they proposed a method that shares KV selection across the dimensions of query, head, and layer, thereby reducing the overhead of KV selection.

**Strengths:**

The sharing-based method proposed in this paper seems novel.

**Weaknesses:**

1.	The experiments only selected 6 out of the 16 commonly used English datasets from Longbench for evaluation, which may not be sufficient to draw solid conclusions. For example, the performance of H2O and Quest on these 6 datasets in Table 3 appears similar, whereas in the Quest paper, H2O showed significantly lower accuracy. This discrepancy may suggest that specific methods perform better on these six datasets. To clarify, the authors are encouraged to provide results across all datasets.

2.	In terms of speedup, the authors analyzed the theoretical complexity of KV cache selection in Table 2. However, this analysis alone is insufficient to demonstrate that the proposed method significantly reduces computational overhead, as the selection cost may only account for a small portion of the entired decoding process. To strengthen this claim, the authors could provide direct evidence of decoding speed in experiments. Additionally, an experimental study on the trade-off between accuracy and speed would be valuable.

3.	The authors provided only a speed comparison with FlashAttention. To better demonstrate the effectiveness of their method, comparisons with other query-aware dynamic token sparsity methods are necessary.

4.	The authors mention using an approximate computation approach from Double Sparse ('Following the approach in Yang et al. (2024)') to calculate approximate attention weights in their implementation. As a result, it is unclear how much of the observed speed gain is attributable to HShare specifically versus the query-aware dynamic token sparsity method itself.

**Questions:**

Please refer to weakness points 1-4

---

> ### Author Response · Authors · 2024-11-20
> **Rebuttal (1/2)**
>
> Thanks for your review and feedback, below is our detailed response.
>
>
> > **W1**: The experiments only selected 6 out of the 16 commonly used English datasets from Longbench for evaluation, which may not be sufficient to draw solid conclusions. For example, the performance of H2O and Quest on these 6 datasets in Table 3 appears similar, whereas in the Quest paper, H2O showed significantly lower accuracy. This discrepancy may suggest that specific methods perform better on these six datasets. To clarify, the authors are encouraged to provide results across all datasets.
>
>
> **Table 1 Evaluation of different methods on sixteen datasets in Longbench.**
> | Method                 | MultiNews | Musique | HotpotQA | Qasper | 2WikiMQA | Repobench-P | TriviaQA | Trec  | Qmsum | NarrativeQA | GovReport | LCC  | Passage-Count | Samsum | Passage-Retrieval-EN | MultifieldQA-EN | Average    |
> |------------------------|------------|---------|----------|--------|----------|-------------|----------|-------|-------|-------------|------------|------|---------------|--------|----------------------|-----------------|------------|
> | streamingllm           | 22.93      | 4.56    | 20.77    | 14.01  | 27.07    | 48.12       | 53.76    | 40.50  | 18.81 | 9.52        | 21.39      | 52.41 | 2.54          | 37.37  | 3.50                  | 20.75           | 24.88  |
> | h2o                    | 22.24      | 6.32    | 25.62    | 14.13  | 25.99    | 47.67       | 60.81    | 44.00    | 19.28 | 11.83       | 21.51      | 57.47 | 2.29          | 38.57  | 3.50                  | 21.10            | 26.40  |
> | quest                  | 25.74      | 6.65    | 23.3     | 17.14  | 28.17    | 44.42       | 81.49    | 61.00    | 20.8  | 14.53       | 25.95      | 53.99 | **4.77**          | 40.67  | 3.50                  | 21.61           | 29.61  |
> | ds                     | **25.98**      | 7.16    | 24.83    | 21.94  | 28.77    | 48.78       | 83.46    | **61.50**  | **20.22** | 15.76       | **26.53**      | **57.75** | 2.21          | **41.48**  | **9.00**                    | **37.55**           | **32.06**    |
> | ours (3/4-3/4-1/2)     | 25.86      | **7.63**    | **24.98**    | **22.13**  | **30.67**    | **49.67**       | **83.92**    | 59.00    | 20.01 | 16.05       | 25.76      | 56.29 | 2.27          | 40.73  | **9.00**                    | 34.46           | 31.78  |
> | ours (1/2-1/2-1/2)     | 24.83      | 7.41    | 25.81    | 21.39  | 29.04    | 48.88       | 83.68    | 57.5  | 20.28 | **16.40**        | 23.88      | 55.89 | 2.5           | 40.13  | 6.00                    | 34.21           | 31.11  |
>
>
> Thanks for the suggestions. The results across the 16 commonly used English datasets from Longbench are shown in Table 1. It can be seen that in the 7/16 dataset, HShare achieves the highest score. On average, HShare outperforms StreamingLLM, H2O, and Quest. Although HShare performs slightly worse than DS, this is a negligible performance decline considering the speed advantage of HShare over DS.
>
> Here we humbly point out that the original H2O paper only applies token sparse attention for the decoding stage, with dense attention used in the prefill stage. In our previous manuscript, we did not take this into account, which may have led to some unfair comparisons, here we apologize for any potential misunderstandings. **In the revised manuscript, for all methods, we apply token-sparse attention to all layers across both the decoding and prefill stages to ensure a fair comparison.**
>
>
> > **W2**: In terms of speedup, the authors analyzed the theoretical complexity of KV cache selection in Table 2. However, this analysis alone is insufficient to demonstrate that the proposed method significantly reduces computational overhead, as the selection cost may only account for a small portion of the entire decoding process. To strengthen this claim, the authors could provide direct evidence of decoding speed in experiments. Additionally, an experimental study on the trade-off between accuracy and speed would be valuable.
>
>
> **Tabel 2 Attention latency ($\downarrow$) and GSM8K accuracy of HShare under different sharing ratios.**
>
> | Sharing Ratio | Attention latency(ms) | GSM8K (flexible/strict)  |
> |---------------|------------|----------------|
> | 1/2           | 0.31    | 0.144/0.136  |
> | 1/4           | 0.23    | 0.135/0.125   |
> | 1/8           | 0.19    | 0.132/0.074 |
> | 1/16          | 0.10     | 0.107/0.032  |
>
> Thank you for the valuable suggestions. Here we provide the attention latency and GSM8K accuracy of our proposed HShare under different sharing ratios in Table 2 to reveal the tradeoff between accuracy and efficiency in our method.
>
> As you mentioned, the theoretical complexity of KV cache selection is insufficient, and a comparison of decoding speed is needed. We have provided system efficiency comparisons with other methods, please refer to the answer to **W3** for further details.

---

> > ### Author Response · Authors · 2024-11-20
> > **Rebuttal (2/2)**
> >
> > > **W3**: The authors provided only a speed comparison with FlashAttention. To better demonstrate the effectiveness of their method, comparisons with other query-aware dynamic token sparsity methods are necessary.
> >
> >
> >
> > **Table 3 Attention latency ($\downarrow$) of different methods across various batch sizes and sequence lengths.**
> >
> >
> > | bs  | seq_len | Flash (dense) | StreamingLLM | H2O  | Quest | DS   | Ours (3/4-3/4-1/2) | Ours (1/2-1/2-1/2) |
> > | --- | ------- | ------------- | ------------ | ---- | ----- | ---- | ------------------ | ------------------ |
> > | 8   | 1k      | 0.23          | 0.03         | 0.09 | 0.20  | 0.14 | 0.12               | 0.09               |
> > | 8   | 2k      | 0.83          | 0.04         | 0.10 | 0.46  | 0.24 | 0.16               | 0.12               |
> > | 8   | 4k      | 1.63          | 0.42         | 0.47 | 0.85  | 0.73 | 0.57               | 0.53               |
> > | 16  | 1k      | 0.44          | 0.03         | 0.09 | 0.28  | 0.13 | 0.12               | 0.09               |
> > | 16  | 2k      | 1.63          | 0.07         | 0.11 | 0.77  | 0.42 | 0.23               | 0.19               |
> > | 16  | 4k      | 3.23          | 0.80         | 0.85 | 2.21  | 1.35 | 1.04               | 0.99               |
> >
> >
> >
> > **Table 4 End-to-end throughput ($\uparrow$) of different methods across various batch sizes and sequence lengths.**
> >
> > | bs  | seq_len | Flash (dense) | StreamingLLM | H2O | Quest | DS  | Ours (3/4-3/4-1/2) | Ours (1/2-1/2-1/2) |
> > | --- | ------- | ------------- | ------ | --- | ----- | --- | ------------------ | ------------------ |
> > | 8   | 1k      | 228           | 264    | 240 | 228   | 228 | 231                | 235                |
> > | 8   | 2k      | 188           | 252    | 234 | 206   | 213 | 222                | 226                |
> > | 8   | 4k      | 118           | 243    | 228 | 152   | 201 | 214                | 217                |
> > | 16  | 1k      | 374           | 465    | 441 | 410   | 423 | 430                | 439                |
> > | 16  | 2k      | 233           | 452    | 416 | 287   | 360 | 398                | 411                |
> > | 16  | 4k      | 136           | 422    | 396 | 175   | 286 | 350                | 365                |
> >
> >
> >
> > Thank you for the nice and constructive suggestions. Table 3 (attention latency) and Table 4 (end-to-end throughput) show a comparison between our method and other query-aware dynamic token sparsity methods, including dynamic eviction algorithm H2O, and the dynamic selection algorithm Quest and DS. The kernel for token sparse attention are designed using OpenAI Triton and the end-to-end speedup evaluation is based on GPT-fast, with the full attention module being replaced by the corresponding token sparsity self-attention module.
> >
> > From the results, we observe that the dynamic eviction algorithm H2O achieves the best efficiency, but it falls short in terms of accuracy. The proposed HShare method delivers a significant speedup compared to other query-aware dynamic token sparsity approaches. For example, in terms of attention latency when batch size is 16 and sequence length is 2k, our method delivers up to a 2.80x speedup over Quest and a 2.21x speedup over DS.
> >
> >
> >
> > > **W4**: The authors mention using an approximate computation approach from Double Sparse ('Following the approach in Yang et al. (2024)') to calculate approximate attention weights in their implementation. As a result, it is unclear how much of the observed speed gain is attributable to HShare specifically versus the query-aware dynamic token sparsity method itself.
> >
> >
> > Thank you for your valuable suggestions. We provide a system comparison in Table3 and Table4. As mentioned above, when batch size is 16 and sequence length is 2k, our algorithm achieves up to 2.21x speedup over DS in terms of attention. We can also find that HShare improve e2e throughput by 1.14x compared to DS in this configuration.
> > In fact, DS proposes to select critical tokens through calculating appriximate attention weights while our algorithm reduces the overhead caused by the token selection algorithm by sharing critical token indices, therefore achieves further improvement.
> >
> >
> > We have added all the above description and experiment into the revised manuscript  (Appendix).
> >
> > ---
> >
> > We hope our responses and modifications could ease your concerns. If you have any other questions, we are glad to provide further responses. We would sincerely appreciate it if you could reconsider your rating and wish to receive your feedback soon.

---

> > > ### Comment · Reviewer_ZRXS · 2024-11-22
> > >
> > > Thanks for your response. However, I still have the following concerns:
> > >
> > > 1. Experiment Settings: The main baselines, Quest DS and Hshare, focus on utilizing dynamic sparsity for decoding acceleration. Upon carefully reviewing the objectives of this work, I confirmed that Hshare also aims to accelerate the decoding stage, as evidenced by the statement in the introduction: "To ensure the sharing accuracy, we compute the corresponding configuration online for each batch of samples after the prefilling phase; the configuration is then applied to the entire decoding phase. This online calculation introduces only a minor increase in the prefill phase's time without incurring any additional overhead during the decode phase"  Why, then, is the sparse attention method employed during the prefill stage in this work's evaluation? Does this introduce unnecessary variables, potentially confounding the assessment of decoding acceleration?
> > >
> > > 2. Performance of Original Attention in settings: In the evaluations, does "Original" refer to lossless vanilla attention, or does it also incorporate prefill sparsity? If prefill sparsity is used, can it truly represent lossless performance? If no, does it imply that the performance gap with existing baselines is partly due to the presence of prefill sparsity?
> > >
> > > 3. LongBench Evaluation: In the rebuttal-phase evaluations on all LongBench datasets, Hshare seems to fall behind DS. For example, in the evaluation on six LongBench datasets provided in the paper, Hshare (3/4-3/5-1/2) achieves a score of 44.74, outperforming DS's 44.54. However, across all 16 datasets, Hshare lags behind DS with an average score of 31.78 compared to 32.06. The authors should clarify the rationale for selecting the six datasets for the main experiments. Otherwise, the average scores across all 16 datasets should be reported in the experimental section.
> > >
> > > 4. Marginal Advantage of Hshare over DS: Hshare’s performance advantage over DS appears limited. Across all 16 datasets, Hshare achieves a score of 31.78, falling behind DS’s 32.06. While the 0.28 gap might seem small, it could be significant in practice, especially considering the substantial resources often required to achieve even minor accuracy improvements in near-lossless compression scenarios. This diminishing returns effect is commonly observed in related works, such as Quest. Therefore, the current comparison, especially when Hshare trades accuracy for speed, is not so convincing. I recommend the authors adjust the share ratio to bring Hshare's accuracy closer to DS (e.g., reducing the gap to below 0.05, or even smaller) and then re-evaluate the speed difference under these conditions. This would provide a more fair comparison.
> > >
> > > 5. Dependence on DS Optimization: I am concerned that Hshare's improvement over Quest primarily stems from leveraging DS's optimizations. The experiments clearly show that DS achieves significant improvements over Quest in both accuracy (29.61 → 32.06) and throughput (175 → 286). However, despite leveraging DS's approximate attention technique, Hshare does not demonstrate a clear advantage over DS. Therefore, I am not convinced that Hshare achieves a comprehensive performance improvement.

---

> > > > ### Author Response · Authors · 2024-11-23
> > > > **Response by authors (1/2)**
> > > >
> > > > Thanks for the response, we truly appreciate the time and effort you've dedicated to reviewing our work. To clarify potential misunderstandings that might affect the evaluation, we first restate the motivation and the position of our work.
> > > >
> > > > **Motivation**: On one hand, among token sparsity methods, static token sparsity approaches generally perform poorly, while query-aware token sparsity methods show relatively better results. However, existing methods (such as Quest and DS) necessarily require selecting critical tokens for each query, and this selection process accounts for a significant portion of the computation time in the attention mechanism. On the other hand, we have observed a high degree of similarity in critical tokens across different layers, heads, and adjacent queries. Based on this observation, we aim to reduce the overhead associated with selecting critical tokens by sharing critical token indices, thus achieving improved performance while preserving accuracy.
> > > >
> > > > **Position of Our Work**: It is important to note that our approach focuses not on the selection of critical tokens, but on the hierarchical sharing of the selected critical token indices. This distinction makes our method fundamentally orthogonal to any existing token sparsity approaches. While our current implementation incorporates certain optimizations from DS, our method is inherently generalizable and can be seamlessly extended to other techniques, highlighting its potential for broader future applications.
> > > >
> > > > In the following response, we provide detailed answers to all the questions point-by-point.
> > > >
> > > > > **Q1**: Experiment Settings.
> > > >
> > > > (1) Yes, we also apply sparsity during the prefill stage. As mentioned in the paper, the sparsity strategy involves selecting initial tokens, critical tokens, and a local window.
> > > >
> > > > (2) We have observed that the similarity patterns across layers and heads remain consistent between the prefill and decode stages for each sample. Consequently, after each prefill stage, HShare computes the sharing configuration, which is determined based on the similarity of critical token indices exhibited across layers and heads during the prefill stage. This calculated sharing configuration is then applied throughout the entire subsequent decode stage. Importantly, this approach does not introduce any additional variables during the decode stage and does not impact the speed evaluation of the decode phase.
> > > >
> > > > > **Q2**: Performance of Original Attention in settings.
> > > >
> > > > (1) Original Attention refers to using Vanilla Attention with FlashAttentionV2 without applying sparsity during either the prefill or decode stages. We apologize for any confusion caused by the lack of clarification and will make this clearer in the paper.
> > > >
> > > > (2) Yes, applying sparsity during the prefill stage is indeed one of the reasons for the performance degradation of all token sparsity methods. In fact, prefill sparsity has a negative impact on our method as well.
> > > >
> > > > > **Q3**: LongBench Evaluation.
> > > >
> > > > Longbench encompasses a variety of task types (e.g., summarization, single-document QA, etc.). The six datasets are chosen as representatives for each task type, as outlined in our paper. Due to space limitations in the main text, we have included the results for all 16 datasets in the appendix. Following the reviewer’s suggestions, we will revise the main experimental section and include a discussion of the average scores.
> > > >
> > > >
> > > > > **Q4**: Marginal Advantage of Hshare over DS.
> > > >
> > > > (1) Regarding the 0.28 gap mentioned by the reviewer, we found that this discrepancy primarily stems from a 3-point loss on the *multifieldqa-en* dataset. For the remaining 15+2 tasks (including GSM8K and CoQA), however, HShare's performance is nearly on par with DS. This indicates that, for the vast majority of tasks, the current sharing ratio is a reasonably effective choice, with performance degradation occurring only in a few specific cases.
> > > >
> > > > To the best of our knowledge, HShare is the first work to introduce the concept of sharing critical KV cache token indices. At present, the selection of the sharing ratio remains relatively heuristic. We conducted an initial investigation into adopting different sharing ratios for different tasks; please refer to Table 1 below. We leave the exploration of dynamically determining whether to share and how to select an appropriate sharing ratio for different tasks as part of our future work.
> > > >
> > > > (2) Additionally, we fully agree with the diminishing returns effect highlighted by the reviewer, which suggests that maintaining the final fraction of accuracy often demands a significant amount of computational resources. We believe this underscores the necessity of making trade-offs. As mentioned earlier, for the vast majority of tasks, our current sharing ratio effectively preserves accuracy, demonstrating the rationale behind the trade-off in most scenarios

---

> > > > > ### Author Response · Authors · 2024-11-23
> > > > > **Response by authors (2/2)**
> > > > >
> > > > > **Table 1 Performance of HShare on different tasks with different sharing ratios under a sparsity of 1/8.**
> > > > > | sharing ratio | GSM8K  (flexible/strict)     | Triviaqa |
> > > > > |-------|-------------|----------|
> > > > > | 1     | 0.167/0.167 | 83.82    |
> > > > > | 1/4   | 0.135/0.125 | 83.68    |
> > > > > | 1/8   | 0.132/0.074 | 83.68    |
> > > > > | 1/16  | 0.107/0.032 | 83.51    |
> > > > > | 1/32  | 0.083/0.017 | 83.09    |
> > > > > | 1/64  | 0.076/0.012 | 82.28    |
> > > > >
> > > > > **Description of Table 1**: GSM8K is a more challenging mathematical reasoning task, while TriviaQA is a simpler question-answering task. We can find that with the increase in the degree of sharing, the performance of GSM8K declines noticeably. When the sharing ratio reaches 1/32, accuracy drops to about 50% of its original value, whereas on TriviaQA, the decline is almost negligible. Therefore, we intuitively believe that for simpler, text-based tasks, a higher sharing ratio can be used to achieve greater acceleration, while for more difficult, reasoning-based tasks, a more moderate sharing ratio should be applied.
> > > > >
> > > > >
> > > > > > **Q5**: Dependence on DS Optimization.
> > > > >
> > > > > With bs = 8 and seq = (1k, 2k, 4k), DS achieves an average throughput increase of 9.56% compared to Quest, while HShare achieves a 5.61% increase compared to DS.
> > > > > With bs = 16 and seq = (1k, 2k, 4k), DS achieves an average throughput increase of 21.67% compared to Quest, while HShare achieves a 14.51% increase compared to DS.
> > > > >
> > > > > Here, we observe an improvement in efficiency, which becomes even more pronounced in scenarios with larger batch sizes and longer sequences. While the improvement we achieve over DS is not as significant as DS's improvement over Quest, this can be partially attributed to the diminishing returns effect mentioned by the reviewer. Moreover, as discussed earlier, considering the technical orthogonality between our approach and methods like DS, as well as the future potential of our method, we believe that our work makes a valuable contribution to the community.
> > > > >
> > > > >
> > > > > ---
> > > > >
> > > > >
> > > > >
> > > > > We hope our detailed responses could answer your questions and ease your concerns. If you have any other questions, we are glad to provide further responses. We would sincerely appreciate it if you could reconsider your rating and wish to receive your feedback soon.

---

> > > > > > ### Comment · Reviewer_ZRXS · 2024-11-25
> > > > > >
> > > > > > I fully understand that HShare is orthogonal to existing methods, and this is precisely why explicit evidence of its effectiveness is essential. Given that HShare builds directly upon the foundation laid by DS and fully incorporates its core contributions, it becomes particularly critical to clearly evaluate the added value of your approach.
> > > > > >
> > > > > > ## Clarification Needed on Prefilling Sparsity in Evaluation
> > > > > >
> > > > > > The inclusion of prefilling sparsity in the evaluation settings for decoding sparsity methods raises significant concerns. This approach diverges from established practices in prior works, such as Quest, DS, and concurrent studies like MagicPIG, none of which employ prefilling sparsity during evaluation. Compounding this issue is the fact that HShare relies on results from the prefilling phase to determine its share configuration. I strongly recommend providing a clear justification for the inclusion of the sparsity in the evaluation. Furthermore, it is crucial to address whether HShare’s strategy would remain effective if dense computation were used during the prefilling stage.
> > > > > >
> > > > > > ## Relation to DS
> > > > > >
> > > > > > It is important to emphasize that claiming a significant speed advantage over DS is inappropriate when HShare demonstrates lower accuracy on key benchmarks like LongBench. Additionally, you acknowledge that further accuracy improvements would require substantial additional resources with diminishing returns. This raises questions about whether HShare can achieve a meaningful speedup over DS under these constraints. **Moreover, authors' arguements for excluding the MultifieldQA dataset (where HShare underperforms by three points) to achieve a comparable average accuracy is problematic. Such selective exclusions introduce bias. For example, if the dataset where HShare performs best, 2WikiMQA (where it leads by two points), were similarly excluded, HShare would still fall behind overall. This compromises the fairness and comprehensiveness of the evaluation.**
> > > > > >
> > > > > > ## Potentially Misleading Claims to Reviewers
> > > > > >
> > > > > > I am concerned that some of the earlier claims may have misled reviewers. Specifically, the statement in the LongBench evaluation that “Notably, with a sharing ratio of 3/4-3/4-1/2, HShare outperforms the existing state-of-the-art methods on average (score)” is misleading. In reality, HShare does not surpass the existing SOTA method, DS, in terms of accuracy, which calls into question the validity of this claim.

---

> > > > > > > ### Author Response · Authors · 2024-11-25
> > > > > > >
> > > > > > > Thanks for the response, we truly appreciate the time and effort you've dedicated to reviewing our work.
> > > > > > >
> > > > > > > > **W1**: Clarification Needed on Prefilling Sparsity in Evaluation
> > > > > > >
> > > > > > > DS, Quest, and StreamingLLM can be classified into sparse attention methods, which could be applied in both the prefilling and decoding stages. We humbly point out that DS does apply sparse attention to its prefill stage, and the source code is shown in the following link. (https://github.com/andy-yang-1/DoubleSparse/blob/main/evaluation/modify_llama.py, https://github.com/andy-yang-1/DoubleSparse/blob/main/evaluation/streaming_llama.py).
> > > > > > > We have included sparse attention in the prefill stage simply in line with that in DS.
> > > > > > >
> > > > > > >
> > > > > > > Regarding the concern that HShare relies on results from the prefilling phase to determine its share configuration, as mentioned earlier, we have observed that the similarity patterns across layers and heads remain consistent between the prefilling and decoding stages for each sample. In other words, it makes no difference whether the sharing configuration is calculated during the prefilling or the decoding phase. Furthermore, regardless of whether sparse attention is applied during the prefill stage, we can compute the sharing configuration using the same approach.
> > > > > > >
> > > > > > > **Thanks for the suggestion. To demonstrate the effectiveness of HShare, we will provide results for all methods with no sparsity applied during the prefilling before the rebuttal deadline.**
> > > > > > >
> > > > > > >
> > > > > > > > **W2**: Relation to DS.
> > > > > > >
> > > > > > > We did acknowledge that HShare achieved a meaningful speedup over DS with a 0.28 average accuracy drop. And we still think it is reasonable since we are doing a tradeoff. Here, taking the reviewer’s suggestion into account, **we are carrying out the related experiments (keep the overall accuracy consistent of DS within 0.05), to see how much acceleration that HShare can achieve. We will provide the corresponding results within the remaining time for the rebuttal.**
> > > > > > >
> > > > > > >
> > > > > > > We apologize for any misunderstandings caused by our response. We did not intend to exclude any dataset to claim our results. In fact, we have mentioned in the main text that our method performs slightly worse than DS on an average of 16 datasets in Sec5.1.3 in the last manuscript. In the last response, we want to claim that different tasks could have different sharing ratio configures, the sharing ratio we have selected is relatively suitable for the vast majority of tasks, and a slight loss on certain datasets is within expectation.
> > > > > > >
> > > > > > > > **W3**: Potentially Misleading Claims to Reviewers
> > > > > > >
> > > > > > > Thank you for the valuable suggestions. To avoid any potential misunderstandings, we have made the related claims clearer in the new manuscript. All revisions and additions are marked in blue.

---

> > > > > > > > ### Comment · Reviewer_ZRXS · 2024-11-26
> > > > > > > >
> > > > > > > > Thanks for your response. However, I did not find any mention of prefilling sparsity in the DS paper. I double-checked the code DS used for its LongBench evaluation. :
> > > > > > > > https://github.com/andy-yang-1/DoubleSparse/blob/main/LongBench/modify_llama.py, lines 100–111:
> > > > > > > >
> > > > > > > > ```python
> > > > > > > > if q_len > 1:
> > > > > > > >     return self.flash_forward(
> > > > > > > >         hidden_states=hidden_states,
> > > > > > > >         attention_mask=attention_mask,
> > > > > > > >         position_ids=position_ids,
> > > > > > > >         past_key_value=past_key_value,
> > > > > > > >         output_attentions=output_attentions,
> > > > > > > >         use_cache=use_cache,
> > > > > > > >         cache_position=cache_position,
> > > > > > > >         position_embeddings=position_embeddings,
> > > > > > > >         **kwargs,
> > > > > > > >     )
> > > > > > > > ```
> > > > > > > > This code shows that dense attention operations are performed directly during the prefilling stage, suggesting that DS's evaluation on LongBench does not appear to employ prefilling sparsity.
> > > > > > > >
> > > > > > > > I would like to emphasize that adopting prefilling sparsity in experimental setups is not a common practice. This can be further substantiated by more related works on decoding acceleration, such as *ShadowKV*[1] and *RetrievalAttention*[2], neither of which introduce unnecessary prefill sparsity in their evaluations.
> > > > > > > >
> > > > > > > > The author's response also suggests that prefilling sparsity is unnecessary for experimental evaluations. Considering that Hshare's configuration is based on prefill results, I am willing to wait for the author's further evaluation results.
> > > > > > > >
> > > > > > > > [1] Sun, Hanshi, et al. "ShadowKV: KV Cache in Shadows for High-Throughput Long-Context LLM Inference." arXiv preprint arXiv:2410.21465 (2024).
> > > > > > > > [2] Liu, Di, et al. "Retrievalattention: Accelerating long-context llm inference via vector retrieval." arXiv preprint arXiv:2409.10516 (2024).

---

> > > > > > > > > ### Author Response · Authors · 2024-11-27
> > > > > > > > > **Response by authors (1/2)**
> > > > > > > > >
> > > > > > > > > Thanks for your patience and waiting, we truly appreciate the time and effort you've dedicated to reviewing our work. According to your suggestions, we carried out some new experiments. The related results and discussion are shown as follows.
> > > > > > > > >
> > > > > > > > >
> > > > > > > > > **About the effectiveness and efficiency of HShare**
> > > > > > > > >
> > > > > > > > > Taking the reviewer’s suggestion into account, we added a new sharing configuration,i.e., (7/8-3/4-1/2) from the configuration of (3/4-3/4-1/2).
> > > > > > > > >
> > > > > > > > > **Table 1 Comparison of DS and HShare on sixteen datasets in Longbench.**
> > > > > > > > >
> > > > > > > > > | Method            | MultiNews | Musique  | HotpotQA  | Qasper | 2WikimQA  | Repobench-P | TriviaQA | Trec  | Qmsum | NarrativeQA | GovReport | LCC   | Passage-Count | Samsum | Passage-Retrieval-EN | MultifieldQA-EN | Average |
> > > > > > > > > |-------------------|-----------|----------|-----------|--------|-----------|-------------|----------|-------|-------|-------------|-----------|-------|---------------|--------|----------------------|-----------------|---------|
> > > > > > > > > | DS                | 25.98     | 7.16     | 24.83     | 21.94 | 28.77     | 48.78       | 83.46    | **61.5**  | 20.22 | 15.76       | **26.53**     | **57.75** | 2.21          | 41.48  | 9.0                  | **37.55**           | 32.06   |
> > > > > > > > > | $\color{blue}{Ours(7/8-3/4-1/2)}$ | $\color{blue}{\textbf{26.30}}$ | $\color{blue}{\textbf{8.15}}$ | $\color{blue}{25.13}$ | $\color{blue}{21.14}$ | $\color{blue}{29.70}$ | $\color{blue}{\textbf{50.22}}$ | $\color{blue}{84.03}$ | $\color{blue}{59.0}$  |$\color{blue}{\textbf{20.40}}$|  $\color{blue}{15.29}$ |  $\color{blue}{25.87}$ | $\color{blue}{57.20}$ |  $\color{blue}{2.27}$ | $\color{blue}{\textbf{42.06}}$  | $\color{blue}{\textbf{9.5}}$  |  $\color{blue}{36.80}$  |  $\color{blue}{\textbf{32.07}}$  |
> > > > > > > > > | ours (3/4-3/4-1/2)     | 25.86      | 7.63    | 24.98    | **22.13**  | **30.67**    | 49.67       | 83.92    | 59.0    | 20.01 | 16.05       | 25.76      | 56.29 | 2.27          | 40.73  | 9.0                    | 34.46           | 31.78  |
> > > > > > > > > | ours (1/2-1/2-1/2)     | 24.83      | 7.41    | **25.81**    | 21.39  | 29.04    | 48.88       | 83.68    | 57.5  | 20.28 | **16.40**        | 23.88      | 55.89 | **2.50**           | 40.13  | 6.0                    | 34.21           | 31.11  |
> > > > > > > > >
> > > > > > > > > **Table 2 End-to-end throughput ($\uparrow$) of DS and HShare across various batch sizes and sequence lengths.**
> > > > > > > > > | bs  | seq_len | DS  | $\color{blue}{Ours(7/8-3/4-1/2)}$ | Ours (3/4-3/4-1/2) | Ours (1/2-1/2-1/2) |
> > > > > > > > > |:---:|:------:|:---:|:------------------------:|:------------------:|:------------------:|
> > > > > > > > > | 8   | 1k     | 228 | $\color{blue}{230}$ | 231                | 235                |
> > > > > > > > > | 8   | 2k     | 213 |  $\color{blue}{220}$ | 222                | 226                |
> > > > > > > > > | 8   | 4k     | 201 | $\color{blue}{212}$ | 214                | 217                |
> > > > > > > > > | 16  | 1k     | 423 |  $\color{blue}{428}$ | 430                | 439                |
> > > > > > > > > | 16  | 2k     | 360 |  $\color{blue}{393}$ | 398                | 411                |
> > > > > > > > > | 16  | 4k     | 286 | $\color{blue}{336}$| 350                | 365
> > > > > > > > >
> > > > > > > > >
> > > > > > > > > Table 1 presents the results of DS and HShare across the 16 commonly used English datasets from Longbench. It can be seen that under the new sharing ratio ($\color{blue}{marked}$ $\color{blue}{by}$ $\color{blue}{the}$ $\color{blue}{blue}$ $\color{blue}{color}$), HShare achieves the highest scores in the most sub-datasets, it achieves competitive performance with DS on average. Table 2. presents the e2e throughput of DS and HShare across various batch sizes and sequence lengths. According to Table 2, we can find that the speed of the new configuration ($\color{blue}{marked}$ $\color{blue}{by}$ $\color{blue}{the}$ $\color{blue}{blue}$ $\color{blue}{color}$)  remains superior to that in DS. For instance, when batch size is 16 and the sequence length is 2k, HShare is 9.17% faster than DS. Notably, as the sequence length increases, the advantage of HShare will be further amplified. For example, when batch size is 16 and the sequence length is 4k, HShare is 17.48% faster than DS.
> > > > > > > > >
> > > > > > > > > According to the results presented in Tables 1 and 2, HShare demonstrates an advantage over DS under a fair comparison. In fact, this conclusion aligns with our earlier discussion, where we noted that the sharing ratio of HShare is a hyperparameter that could influence the balance between effectiveness and efficiency.
> > > > > > > > >
> > > > > > > > > Thank you for your valuable suggestions. We have incorporated the aforementioned description and experiments into the revised manuscript.

---

> > > > > > > > > > ### Author Response · Authors · 2024-11-27
> > > > > > > > > > **Response by authors (2/2)**
> > > > > > > > > >
> > > > > > > > > > **About the sparsity in prefilling stage**
> > > > > > > > > >
> > > > > > > > > >
> > > > > > > > > >
> > > > > > > > > > First, we humbly point out that the authors of DS **released the LongBench code just two weeks ago**, see commit history https://github.com/andy-yang-1/DoubleSparse/commits/main/ for details.  However, HShare was initially completed in September and then submitted to ICLR 2025. At that time, DS only had the following scripts: https://github.com/andy-yang-1/DoubleSparse/blob/main/evaluation/modify_llama.py, https://github.com/andy-yang-1/DoubleSparse/blob/main/evaluation/streaming_llama.py, and we followed the original scripts. We apologize for not noticing their latest release, and we truly appreciate you bringing this to our attention.
> > > > > > > > > >
> > > > > > > > > >
> > > > > > > > > >
> > > > > > > > > > **Table 3 Validation of the impact of not applying sparsity during the prefilling stage on sixteen datasets in Longbench.**
> > > > > > > > > > | Method | MultiNews | Musique | HotpotQA | Qasper | 2WikimQA | Repobench-P | TriviaQA | Trec | Qmsum | NarrativeQA | GovReport | LCC | Passage-Count | Samsum | Passage-Retrieval-EN | MultifieldQA-EN | Average |
> > > > > > > > > > |:------:|:--------:|:-------:|:--------:|:------:|:--------:|:-----------:|:--------:|:----:|:------:|:-----------:|:----------:|:---:|:-------------:|:------:|:-------------------:|:--------------:|:-------:|
> > > > > > > > > > | StreamingLLM | 23.34 | 5.79 | 23.89 | 13.80 | 29.79 | 48.06 | 80.82 | 54.5 | 19.09 | 13.62 | 20.64 | 55.18 | 2.02 | 37.87 | 4.0 | 23.69 | 28.51 |
> > > > > > > > > > | H2O | 24.88 | 6.24 | 27.58 | 17.32 | 31.38 | **51.81** | 81.90 | 62.5 | 20.74 | 17.01 | 23.44 | 56.55 | 1.87 | 40.18 | 5.0 | 33.64 | 31.38 |
> > > > > > > > > > | Quest | **26.49** | 4.62 | 21.94 | 16.96 | 29.42 | 50.24 | 82.42 | **64.0** | **21.36** | **18.21** | 25.46 | **56.74** | 2.33 | 40.91 | **6.5** | 31.99 | 31.35 |
> > > > > > > > > > | DS | 26.30 | **8.72** | 27.44 | **20.98** | **32.01** | 48.44 | 83.23 | 61.0 | 20.75 | 17.54 | **27.04** | 54.87 | 2.33 | **41.30** | 6.0 | 35.83 | 32.11 |
> > > > > > > > > > | Ours(7/8-3/4-1/2) |  26.22 |  7.52 |  27.29 | 19.98|  31.74 |51.33 | **83.91** | 60.5 |20.77| 17.15 |26.63 |55.18 |2.98 | 40.66 |**6.5** | **36.65** |**32.13** |
> > > > > > > > > > | Ours(3/4-3/4-1/2) | 26.05 | 8.42 | 27.32 | 20.12 | 31.46 | 50.79 | **83.91** | 60.5 | 20.59 | 17.31 | 25.85 | 56.23 | **3.15** | 39.77 | 6.0 | 34.08 | 31.97 |
> > > > > > > > > > | Ours(1/2-1/2-1/2) | 25.22 | 8.09 | **29.13** | 19.60 | 31.43 | 51.63 | 82.16 | 58.5 | 20.74 | 16.66 | 23.93 | 56.11 | 2.30 | 40.02 | 6.0 | 32.37 | 31.49 |
> > > > > > > > > >
> > > > > > > > > >
> > > > > > > > > >
> > > > > > > > > > Table 3 presents the results of various methods on LongBench without applying sparsity in the prefilling stage. We can observe that the performance of Ours (7/8-3/4-1/2) still achieves competitive results of DS. Compared to the results where sparsity is applied during the prefill stage, we observe that most methods show a slight improvement while H2O achieves a significant improvement. We suspect this is because H2O performs topK operations based on historical attention scores to achieve sparsity. Therefore, if sparsity is not applied during the prefilling stage, the topK operation during decoding will access comprehensive historical information, resulting in more accurate results and thus improving accuracy.

---

> > > > > > > > > > > ### Comment · Reviewer_ZRXS · 2024-11-29
> > > > > > > > > > >
> > > > > > > > > > > Thank your for updating the results. I would raise points if the aforementioned claims, technical details, and experimental results are addressed and incorporated in the main text, as including them solely or partially on the appendix is insufficient for clarification.

---

> > > > > > > > > > > > ### Author Response · Authors · 2024-11-30
> > > > > > > > > > > >
> > > > > > > > > > > > Thanks for your patience and waiting! We sincerely appreciate the constructive suggestions you provided during the rebuttal period, which have further improved the quality of our paper. Based on your suggestion, we have added the new experimental results, the corresponding technical details, and discussions in the main text. Due to the fact that the last day for modifying the PDF during the rebuttal period is November 27th, and considering the principle of double-blind review, **we temporarily upload the [revised paper](https://anonymous.4open.science/r/ICLR25-2383-0D08/ICLR25-2383.pdf) to (https://anonymous.4open.science/r/ICLR25-2383-0D08/ICLR25-2383.pdf) as anonymous authors. We apologize for the slow loading time of this link and any potential formatting issues with the online preview, thus we recommend downloading the PDF for optimal viewing。**
> > > > > > > > > > > >
> > > > > > > > > > > > Next, we will summarize the modifications.
> > > > > > > > > > > >
> > > > > > > > > > > >
> > > > > > > > > > > > **About the fair comparison.**
> > > > > > > > > > > >
> > > > > > > > > > > > In the revised manuscript, we directly present the results of all 16 English datasets from the LongBench in the main text. We also add the sharing configuration (i.e., 7/8-3/4-1/2) to demonstrate that HShare can achieve competitive accuracy performance with DS while gaining a speed advantage. The related results could be found in Tab.3, Tab.4, and Tab.5 in the revised manuscript.
> > > > > > > > > > > >
> > > > > > > > > > > >
> > > > > > > > > > > > **About the sparsity in the prefill stage.**
> > > > > > > > > > > >
> > > > > > > > > > > > We follow the reviewer's suggestion by maintaining dense attention during the prefill stage and applying token-sparse only in the decoding stage. The results of Tab.3 and Tab.4 are conducted under this setting, and the related technical details as well as discussions are shown in Section 5. Moreover, to further validate the effectiveness of HShare, we present experiments with token-sparse attention applied to both the prefill and decoding stages in Appendix A.1 and Appendix A.3.
> > > > > > > > > > > >
> > > > > > > > > > > >
> > > > > > > > > > > >
> > > > > > > > > > > > **About the trade-off between effectiveness and efficiency.**
> > > > > > > > > > > >
> > > > > > > > > > > > In the revised manuscript, we move the comparison of system efficiency with other baselines from the Appendix to the main text and add more discussion about the trade-off between effectiveness and efficiency. More details are provided in Section 5.2.
> > > > > > > > > > > >
> > > > > > > > > > > >
> > > > > > > > > > > > **About the other suggestions.**
> > > > > > > > > > > >
> > > > > > > > > > > > For your other comments, as well as those from the other reviewers, we have highlighted the corresponding responses in blue. These revisions have been incorporated throughout both the main text and the Appendix.
> > > > > > > > > > > >
> > > > > > > > > > > >
> > > > > > > > > > > > ---
> > > > > > > > > > > > We hope our revised manuscript could address your concerns. If you have any further suggestions for our manuscript, we would be happy to make additional revisions. Thank you again for the time and effort you have dedicated to reviewing our work, and we look forward to receiving your feedback soon.

---

> > > > > > > > > > > > ### Author Response · Authors · 2024-12-03
> > > > > > > > > > > >
> > > > > > > > > > > > Dear Reviewer ZRXS,
> > > > > > > > > > > >
> > > > > > > > > > > > We hope this email finds you well. We understand your busy schedule and sincerely appreciate the valuable feedback you’ve provided. As the discussion phase is nearing its end, we are eager to know if our revisions have met your expectations. If so, we would be rather appreciative if they could merit an elevated score through your evaluation.
> > > > > > > > > > > >
> > > > > > > > > > > > Thanks for your time and effort.
> > > > > > > > > > > >
> > > > > > > > > > > > Best regards, Authors

---

> > > > > > > > > > > > > ### Comment · Reviewer_ZRXS · 2024-12-03
> > > > > > > > > > > > >
> > > > > > > > > > > > > Thank you for providing new results and I would like to raise points. Please make sure the updated results on the anonymous link are show in the revised manuscript.

---

> > > > > > > > > > > > > > ### Author Response · Authors · 2024-12-03
> > > > > > > > > > > > > >
> > > > > > > > > > > > > > Thank you very much for your response and the increased score! We deeply appreciate your participation during the rebuttal period and your constructive feedback. We will definitely incorporate the content from the anonymous link in our final manuscript.

---

> ### Author Response · Authors · 2024-12-02
>
> Dear Reviewer ZRXS,
>
> Thank you for actively participating in the discussions during the rebuttal period and for your constructive feedback. We have made revisions to our manuscript based on your suggestions.
>
> As the rebuttal phase is nearing its end, we look forward to hearing from you, should our modifications and clarifications in the revised manuscript meet your expectations.
>
> We greatly appreciate your time and effort.
>
>
> Best regards,
> Authors

---

### Official Review · Reviewer_mExh · 2024-11-03

**Soundness:** 3
**Presentation:** 3
**Contribution:** 3
**Rating:** 6
**Confidence:** 4

**Summary:**

This paper explores the similarities in the KV cache tokens across neighboring queries, layers, and heads. Building on this finding, the paper introduces an advanced solution named HShare, which leverage a hierarchical framework for KV cache sharing. HShare enables the sharing of critical KV cache token indices across layers, heads, and queries, substantially reducing the computational load associated with query-aware dynamic token sparsity. Additionally, we propose a greedy algorithm that dynamically selects the optimal layer-level and head-level sharing configurations during the decoding phase. Emprical study was conducted to assess HShare's effectiveness and efficiency on various tasks using three models—LLaMA2-7b, LLaMA3-70b, and Mistral-7b. Experimental results reveal that HShare preserves accuracy while enhancing system efficiency.

**Strengths:**

- S1. This paper addresses a crucial and cutting-edge problem in the field, tackling the computational challenges associated with managing KV cache in long-context LLMs.

- S2.  The techique solution is well-desgined. By optimizing the reuse and sharing of critical KV cache token indices across queries, layers, and heads, the proposed solution significantly reduces computational overhead without compromising accuracy.

**Weaknesses:**

- W1. The main concern I have is about the system-efficiency evaluation in Section 5.2. Concretely, I think the baseline is a little weak; we know FlashAttention could be the state-of-the-art implementation, but it is not equipped with advanced algorithm design for selective KV-cache usage. As motivated in the introduction, other methods also attempt to reduce the computation budget based on sparsity; I think the author should report the corresponding system efficiency with the same set of baselines in Section 5.1 to fully reveal the trade-offs introduced in Table 1.

- W2. There are some trivial presentation issues. For example, on page 6, line 270, there is no space between "across" and "layers".

**Questions:**

Please address the corresponding problem listed in the Weaknesses Section.

---

> ### Author Response · Authors · 2024-11-20
> **Rebuttal**
>
> Thank you for acknowledging our work. Your rating has provided us with great encouragement, and your detailed suggestions are really helpful. Below is our response.
>
>
> > **W1**. I think the author should report the corresponding system efficiency with the same set of baselines in Section 5.1 to fully reveal the trade-offs introduced in Table 1.
>
>
> **Table 1 Attention latency ($\downarrow$) of different methods across various batch sizes and sequence lengths.**
> | bs  | seq_len | Flash (dense) | StreamingLLM | H2O  | Quest | DS   | Ours (3/4-3/4-1/2) | Ours (1/2-1/2-1/2) |
> | --- | ------- | ------------- | ------------ | ---- | ----- | ---- | ------------------ | ------------------ |
> | 8   | 1k      | 0.23          | 0.03         | 0.09 | 0.20  | 0.14 | 0.12               | 0.09               |
> | 8   | 2k      | 0.83          | 0.04         | 0.10 | 0.46  | 0.24 | 0.16               | 0.12               |
> | 8   | 4k      | 1.63          | 0.42         | 0.47 | 0.85  | 0.73 | 0.57               | 0.53               |
> | 16  | 1k      | 0.44          | 0.03         | 0.09 | 0.28  | 0.13 | 0.12               | 0.09               |
> | 16  | 2k      | 1.63          | 0.07         | 0.11 | 0.77  | 0.42 | 0.23               | 0.19               |
> | 16  | 4k      | 3.23          | 0.80         | 0.85 | 2.21  | 1.35 | 1.04               | 0.99               |
>
>
>
> **Table 2 End-to-end throughput ($\uparrow$) of different methods across various batch sizes and sequence lengths.**
> | bs  | seq_len | Flash (dense) | StreamingLLM | H2O | Quest | DS  | Ours (3/4-3/4-1/2) | Ours (1/2-1/2-1/2) |
> | --- | ------- | ------------- | ------ | --- | ----- | --- | ------------------ | ------------------ |
> | 8   | 1k      | 228           | 264    | 240 | 228   | 228 | 231                | 235                |
> | 8   | 2k      | 188           | 252    | 234 | 206   | 213 | 222                | 226                |
> | 8   | 4k      | 118           | 243    | 228 | 152   | 201 | 214                | 217                |
> | 16  | 1k      | 374           | 465    | 441 | 410   | 423 | 430                | 439                |
> | 16  | 2k      | 233           | 452    | 416 | 287   | 360 | 398                | 411                |
> | 16  | 4k      | 136           | 422    | 396 | 175   | 286 | 350                | 365                |
>
>
> Thank you for the constructive advice. Here, we provide system efficiency comparisons as the same settings in Section 5.1 (including two KV cache eviction algorithms **StreamingLLM** and **H2O**, two query-aware token sparsity algorithms **Quest** and **DS**). The results are shown in Table 1 (attention latency) and Table 2 (end-to-end throughput).
>
> From the results, it can be observed that KV cache eviction algorithms have a clear advantage in system efficiency compared to query-aware token sparsity algorithms. Specifically, StreamingLLM achieves the fastest speed as it applies a fixed sparsity pattern. However, both StreamingLLM and H2O fall short in terms of accuracy. In contrast, HShare not only preserves accuracy but also achieves significant speedup compared to other query-aware dynamic token sparsity methods. For example, in terms of attention latency when batch size is 16 and sequence length is 2k, it delivers up to a 2.80x speedup over Quest and a 2.21x speedup over DS.
>
> We have added the above description and experiment to the revised manuscript  (Appendix).
>
>
>
> > **W2**. There are some trivial presentation issues. For example, on page 6, line 270, there is no space between "across" and "layers".
>
> Thanks. We have taken note of the issues you mentioned and have made the necessary revisions to the wording in the article. Specifically, the spacing issue between "across" and "layers" on line 270 of page 6 has been corrected. We really appreciate your careful review.
>
>
>
> ---
>
> We hope this response could answer your questions and address your concerns, looking forward to receiving your further feedback soon.

---

> > ### Comment · Reviewer_mExh · 2024-11-23
> >
> > Thank the author for providing additional experimental results! I stay positive about the paper.

---

> > > ### Author Response · Authors · 2024-11-23
> > >
> > > Thank you for recognizing our work and for your timely response!

---

### Author Response · Authors · 2024-11-20
**General Response by Authors**

We express our gratitude to all the reviewers for dedicating their time and providing valuable comments. They acknowledged that our work is meaningful (mExh,Bqih), well-designed (mExh,RqY3,Bqih), contributive (mExh,DdXB,Bqih), effective (RqY3,Bqih), and presents a novel approach (ZRXS, DdXB). While the overall feedback from the reviewers is positive, reviewer ZRXS has some concerns regarding the evaluation of LongBench and the efficiency comparison with other baselines. In the following response, we provide detailed answers to all the questions and comments point-by-point. In particular, we have included results from 16 commonly used English datasets from LongBench, as well as system efficiency comparisons with other methods, in the rebuttal. Additionally, we have revised the manuscript based on the reviewers' suggestions, with all revisions and additions clearly marked in blue.

We deeply appreciate the suggestions to improve this paper. If you have any further questions, please let us know so that we can provide a timely follow-up response.

---

### Meta-Review · Area_Chair_zteG · 2024-12-18

**Metareview:**

Summary:
This paper introduces HShare, a hierarchical framework for sharing KV cache token indices across layers, heads, and queries in LLM inference. The authors exploit observed similarities in critical token usage patterns to reduce computational overhead while maintaining accuracy, proposing a greedy algorithm for determining optimal sharing configurations that demonstrates significant speedups across multiple models and tasks.

Main Strengths:
The technical merit of this work is evident in its hierarchical sharing approach, which is built on theoretical foundations and employs an effective greedy configuration algorithm. The empirical validation is sufficient, featuring evaluation across multiple models and tasks, demonstrating good accuracy maintenance alongside significant speedup, and including ablation studies and comparisons. The practical impact of this work is significant as it addresses an important efficiency challenge in LLM inference, demonstrates compatibility with multiple architectures, and shows clear potential for real-world deployment.

Main Weaknesses:
The initial evaluation had several limitations, including restricted dataset coverage in LongBench evaluation, missing baselines in efficiency comparisons, and inconsistency in the prefill stage sparsity approach. The presentation suffered from clarity issues, with some unclear technical descriptions, language problems in several sections, and missing details on hierarchical integration. There were also practical concerns regarding limited discussion of sharing ratio selection, inadequately explored trade-offs, and questions about broader applicability.

**Additional Comments On Reviewer Discussion:**

Outcomes from Author-Reviewer Discussion:
The authors provided comprehensive responses and significant improvements to their evaluation by adding results for all 16 LongBench datasets, including an additional sharing configuration, and providing efficiency comparisons with more baselines. They clarified experimental settings by explaining prefill/decode stage sparsity settings, adding dense prefill evaluations, and better illustrating trade-offs between accuracy and efficiency. Technical clarity was improved through the addition of a hierarchical framework illustration, clarification of sharing mechanisms, and enhancement of language and technical descriptions.

Reviewer Agreement:
There was consensus among reviewers regarding the technical soundness, novelty, comprehensive empirical evaluation, and practical significance of the work. However, reviewers had mixed views on the extent of accuracy-efficiency trade-offs, comparison fairness with baselines, and presentation clarity.

Suggestions to Improve:
The technical presentation could be enhanced by further clarifying hierarchical integration, better explaining sharing ratio selection, and adding more architectural diagrams. The evaluation would benefit from ablation studies, testing on more varied architectures, and better justification of experimental choices. Practical aspects could be strengthened by discussing deployment considerations, exploring dynamic sharing strategies, and addressing scalability questions.

---

### Decision · Program_Chairs · 2025-01-22

Accept (Poster)